# The effect of hybridization on transposable element accumulation in an undomesticated fungal species

Mathieu Hénault[1,2,3,4]*, Souhir Marsit[1,2,3,4,5], Guillaume Charron[1,3,4,5†], Christian R Landry[1,2,3,4,5]

[1]Institut de Biologie Intégrative et des Systèmes (IBIS), Université Laval, Québec, Canada; [2]Département de biochimie, microbiologie et bioinformatique, Université Laval, Québec, Canada; [3]Quebec Network for Research on Protein Function, Engineering, and Applications (PROTEO), Université Laval, Québec, Canada; [4]Université Laval Big Data Research Center (BDRC_UL), Québec, Canada; [5]Département de biologie, Université Laval, Québec, Canada

**Abstract** Transposable elements (TEs) are mobile genetic elements that can profoundly impact the evolution of genomes and species. A long-standing hypothesis suggests that hybridization could deregulate TEs and trigger their accumulation, although it received mixed support from studies mostly in plants and animals. Here, we tested this hypothesis in fungi using incipient species of the undomesticated yeast *Saccharomyces paradoxus*. Population genomic data revealed no signature of higher transposition in natural hybrids. As we could not rule out the elimination of past transposition increase signatures by natural selection, we performed a laboratory evolution experiment on a panel of artificial hybrids to measure TE accumulation in the near absence of selection. Changes in TE copy numbers were not predicted by the level of evolutionary divergence between the parents of a hybrid genotype. Rather, they were highly dependent on the individual hybrid genotypes, showing that strong genotype-specific deterministic factors govern TE accumulation in yeast hybrids.

*For correspondence:
mathieu.henault.1@ulaval.ca

Present address: †Centre de foresterie des Laurentides, Ressources naturelles Canada, Québec, Canada

## Introduction

Hybridization is increasingly recognized as an important component of species evolution (*Mallet, 2005*; *Mallet et al., 2016*; *Stukenbrock, 2016*). Abundant empirical evidence shows that hybridization frequently leads to transgressive phenotypes in plants, animals, and fungi (*Langdon et al., 2019*; *Rieseberg et al., 1999*). Mutation rates are traits for which transgressive segregation can have a major impact on the evolutionary fate of hybrids, with potential implications for the emergence of reproductive barriers and speciation. Insertions caused by transposable elements (TEs) are a class of mutations that could be particularly responsive to hybridization. TEs are diverse and ubiquitous DNA sequences that can duplicate and spread within genomes, forming families of dispersed repeated sequences (*Bourque et al., 2018*). They can either replicate through 'copy-and-paste' (retrotransposons) or 'cut-and-paste' (DNA transposons) mechanisms. TEs were first identified in cytological studies of chromosome instability in *Zea mays* (*McClintock, 1950*). In her seminal 1984 perspective, *McClintock, 1984* summarized evidence suggesting that TEs respond to various types of 'genome shocks', one of which she suspected was hybridization. Since McClintock's early insights, many studies investigated the association between TEs and hybridization across taxonomic groups and documented increased TE activity in hybrids. We refer to this increase as TE reactivation, but our use of this term implies no mechanistic basis and other terms like derepression are used in the literature. Reactivation was evidenced by increased TE abundance, novel TE insertions, TE

**eLife digest** Hybrids arise when two populations of organisms that are related but genetically different mate and produce offspring. Hybridization has long been regarded as one of the many ways species evolve. Studying the changes in the genome that result from this process can provide insights into evolutionary history and predict the outcome of mixing between genetically different populations. In fact, the inability of two organisms to mate and produce viable and fertile hybrids has been used as a way to define species. It has been speculated that the infertility of many hybrids is due to short sequences of DNA in the genome called transposable elements. These elements are sequences of DNA that, when active, can move to a different position in the genome, causing mutations. It is thought that the process of hybridization may be activating transposable elements leading to the infertility often observed in hybrids.

The activation of transposable elements in hybrids has been studied in animals and plants, and usually, the hybrids studied were either generated in the laboratory or found in the wild. Fungal species, such as the yeast *Saccharomyces paradoxus*, have hundreds of wild strains, including many hybrids, and can also be crossed in the laboratory to produce new hybrids, allowing a combined approach to studying the activation of transposable elements. Hénault et al. used this yeast to investigate whether hybridization leads to increased activity of transposable elements in fungi.

To test this hypothesis, Hénault et al. analyzed the genomes of hundreds of natural strains of *S. paradoxus* to count and locate their transposable elements and establish evolutionary relationships between them. Next, they crossed different strains in the laboratory to see how the transposable elements would act upon hybridization.

If transposable elements were activated by hybridization, then hybrids would accumulate more transposable elements. However, the analyses did not show increased numbers of transposable elements in wild hybrids of *S. paradoxus*. This could be explained by an actual absence of transposable element activation, or by natural selection eliminating individuals that accumulate more transposable elements. To determine which is the case, Hénault et al. next recreated several hybrids in the laboratory and reproduced them for hundreds of generations. Hybrids were grown in the laboratory such that natural selection was almost incapable of favoring some yeasts over others, allowing the hybrids to accumulate transposable elements. These experiments revealed that hybrids accumulated transposable elements at different and largely unpredictable rates. Indeed, closely related hybrids often had highly different numbers of transposable elements in their genomes after being reproduced in the laboratory.

These observations indicate that the accumulation of transposable elements depends on various factors and cannot be easily predicted, and that hybridization may only be a small piece of the puzzle. Additionally, Hénault et al. demonstrated that undomesticated organisms like fungi can provide unique insights into evolutionary hypotheses.

---

epigenetic changes or higher TE transcript levels (*Dion-Côté et al., 2014*; *Kidwell et al., 1977*; *Labrador et al., 1999*; *Liu and Wendel, 2000*; *O'Neill et al., 1998*; *Picard, 1976*; *Ungerer et al., 2006*). However, hybrid reactivation is not universal, as many studies across a similar taxonomic range found no evidence for it (*Coyne, 1989*; *Göbel et al., 2018*; *Hey, 1988*; *Kawakami et al., 2011*). In fungi, a recent preprint showed that no TE reactivation occurs in experimental interspecific hybrids between the yeasts *Saccharomyces cerevisiae* and *Saccharomyces uvarum* (*Smukowski Heil et al., 2020*). Thus, TE reactivation exhibits no clear taxonomic distribution, and the conditions favorable for its manifestation are still unclear.

Studies of hybrids in controlled crosses revealed genetic bases to TE reactivation, but less focus was put on cases of natural hybrids (but see *Kawakami et al., 2011*; *Ungerer et al., 2009*). In addition to the genetic structure of populations, environmental pressures and demographic factors can influence the action of evolutionary forces on TEs, thus impacting the genomic landscapes of TEs in natural populations, including hybrids. Population genetic models have investigated the action of the major evolutionary forces on mean TE copy number (CN) per individual (*Charlesworth and Charlesworth, 1983*; *Charlesworth and Langley, 1989*). Three such forces are deterministic (*Charlesworth et al., 1994*). First, TE CN increases at a certain transposition rate, which

incorporates the various forms of transposition regulation (either by the host or by TEs themselves). Second, TEs are removed from the genome at a certain excision rate. Third, most TE insertions are deleterious, such that individual fitness is assumed to be a decreasing function of CN and natural selection acts to purge TE insertions from populations. The fourth important force is genetic drift, a stochastic force that impacts the efficiency of natural selection by increasing the probability of fixation of deleterious mutations (*Lynch, 2007*). The intensity of drift is inversely proportional to the effective population size $N_e$. In populations with small $N_e$, drift renders natural selection inefficient, such that deleterious mutations are more likely to fix by chance. In principle, any of those evolutionary forces could affect hybrids in a way that differs from their parents, leading to hybrid-specific dynamics in the accumulation of TEs.

Population data in *Drosophila* indicate that deterministic forces have a predominant influence on the allele frequency spectrum of TE insertions (*Charlesworth and Langley, 1989*), consistent with a minimal contribution from drift. Among the deterministic forces, transposition rate is arguably the most likely to be impacted by hybridization, for instance if genetic incompatibilities impair transposition regulation mechanisms. A classic illustration of this is hybrid dysgenesis in *Drosophila*, a syndrome caused by mismatches between paternal TEs and egg-loaded pools of Piwi-interacting RNAs (piRNAs) that lead to inefficient TE repression (*Brennecke et al., 2008*; *Bucheton et al., 1984*; *Erwin et al., 2015*; *Rubin et al., 1982*). Similarly, hybrids between *Arabidopsis thaliana* and *Arabidopsis arenosa* show overexpression of a TE family from the *A. arenosa* subgenome that is dependent on the ploidy level of the parental accessions involved, suggesting that dosage imbalance of epigenetic regulators causes reactivation (*Josefsson et al., 2006*). It can also be expected that the number of genetic incompatibilities increases with genetic divergence, yielding a positive relationship between evolutionary divergence and TE deregulation (*Orr, 1995*; *Parhad and Theurkauf, 2019*). Additionally, evidence from *Drosophila* suggests that hybridization may impact excision rates for specific TE families (*Coyne, 1989*). Along with these deterministic factors, reductions in natural selection efficiency should also be considered. Evidence from *Arabidopsis* (*Lockton et al., 2008*; *Lockton and Gaut, 2010*), *Caenorhabditis* (*Dolgin et al., 2008*) and *Daphnia* (*Ye et al., 2017*) suggests that long-term differences in $N_e$ caused by demography or mating system are associated with variation in TE content (*Lynch and Conery, 2003*). A reduction in $N_e$ can be expected when a novel hybrid lineage is formed. Indeed, it has been hypothesized that population bottlenecks inherent with hybrid speciation in *Helianthus* might have played a role in the accumulation of TEs through a decreased efficiency of purifying selection (*Ungerer et al., 2009*).

An emerging eukaryotic model system in ecological and evolutionary genomics is the yeast *Saccharomyces paradoxus*, the undomesticated sister species of the budding yeast *S. cerevisiae*. *S. paradoxus* has a complex admixture history in North America, making it an ideal model system to test hybridization-related evolutionary hypotheses in natural populations and in the laboratory (*Figure 1*; *Hénault et al., 2017*). North American *S. paradoxus* lineages comprise the Eurasian lineage *SpA*, the endemic incipient species *SpB* and *SpC* (hereafter termed pure lineages), and the admixed lineages *SpC\**, *SpD₁* and *SpD₂* (hereafter termed hybrid lineages; *Figure 2a*). *SpC\** is an incipient hybrid species that arose from a post-glacial secondary contact between the endemic lineages *SpB* and *SpC* and consists of mostly *SpC*-like genomes with ~5% introgressions from *SpB* (*Leducq et al., 2016*), while *SpD₁* and *SpD₂* arose more recently from hybridization between *SpC\** and *SpB* and have roughly equal genomic contributions from both parents (*Eberlein et al., 2019*; *Figure 2b*).

Yeasts of the *Saccharomyces* genus are valuable model species to investigate the interplay between hybridization and transposition. *Saccharomyces* genomes contain many long terminal repeats (LTR) retrotransposons of the Ty1/copia (Ty1, Ty2, Ty4 and Ty5) and Ty3/gypsy (Ty3) families. Ty1/copia and Ty3/gypsy retrotransposon superfamilies (*Wicker et al., 2007*) respectively correspond to the Pseudoviridae and Metaviridae virus families (*Krupovic et al., 2018*) and have a broad taxonomic distribution. Ty elements comprise two overlapping open reading frames (ORFs) within a ~5 kb internal sequence, which is flanked by two ~300 bp LTRs in direct orientation. The two LTRs of full-length elements are prone to intra-element recombination, leaving behind solo LTRs in the genome (*Parket et al., 1995*). Ty1 in the budding yeast *S. cerevisiae* is among the best characterized eukaryotic TEs, benefiting from decades of research on its molecular and cellular biology and, more recently, on its population genomics (*Bleykasten-Grosshans et al., 2013*; *Curcio et al., 2015*). Although *Saccharomyces* genomes harbor a relatively modest fraction of Ty elements (~1–3% in

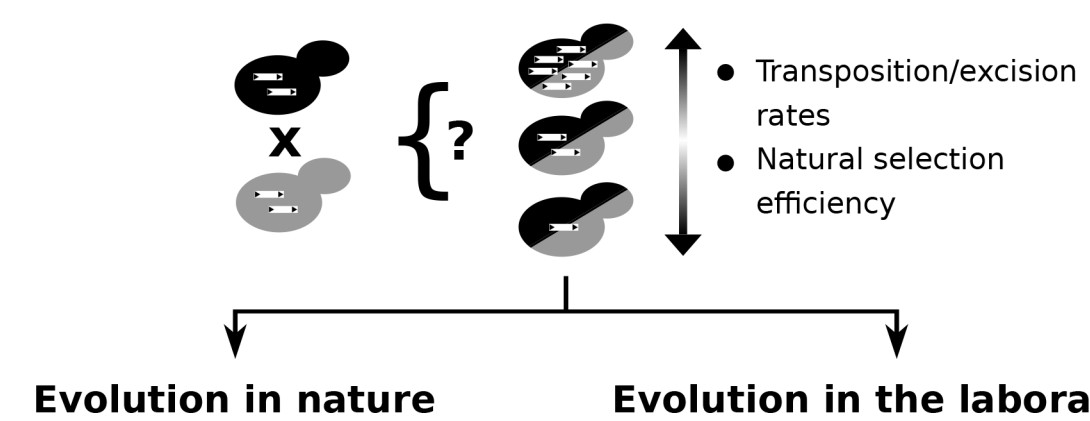
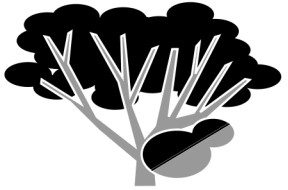
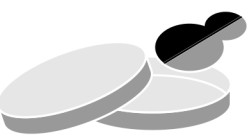

- Transposition/excision rates
- Natural selection efficiency

## Evolution in nature

- 6 long-read based whole-genome assemblies of wild strains
- 207 short-read libraries of wild strains

  - Population structure
  - Environmental pressures
  - Natural selection efficiency

## Evolution in the laboratory

- 744 short-read libraries of 372 experimental lines evolved under relaxed natural selection

  - Evolutionary divergence between parents
  - Combined TE content of parents

**Figure 1.** Population genomic data and laboratory evolution experiments allow to uncover the effects of various factors on transposable element (TE) accumulation in hybrids. Upon hybridization, the TE content of a hybrid is determined by equal contributions from the TE contents of its parents. Subsequent evolution can either leave the hybrid TE content unchanged, or drive it to increase or decrease. These changes can be driven by biases in transposition or excision rates or by variation in natural selection efficiency. Two complementary approaches can be used to investigate which factors drive TE dynamics in hybrids. First, genomic data from natural populations can be harnessed to understand how population structure, environmental pressures and natural selection efficiency shaped natural variation in TE content. Second, evolution experiments in laboratory controlled conditions (constant environment and relaxed natural selection) allow to test the effect of properties of artificial hybrid genotypes on TE accumulation, namely evolutionary divergence between parents and initial TE content.

size), their streamlined genomes (~12 Mbp) and short generation time (~2 hr) provide great power to perform TE comparative genomics and laboratory evolution experiments.

Here, we use population genomic data and laboratory evolution experiments on *S. paradoxus* and its sibling species *S. cerevisiae* to investigate the hybrid reactivation hypothesis and, more generally, the factors governing TE accumulation in natural and experimental lineages. Our results show that natural and experimental hybrids exhibit no reactivation of TEs. Rather, we show that deterministic factors like population structure and the properties of individual hybrid genotypes are major determinants of TE content evolution in hybrid genomes.

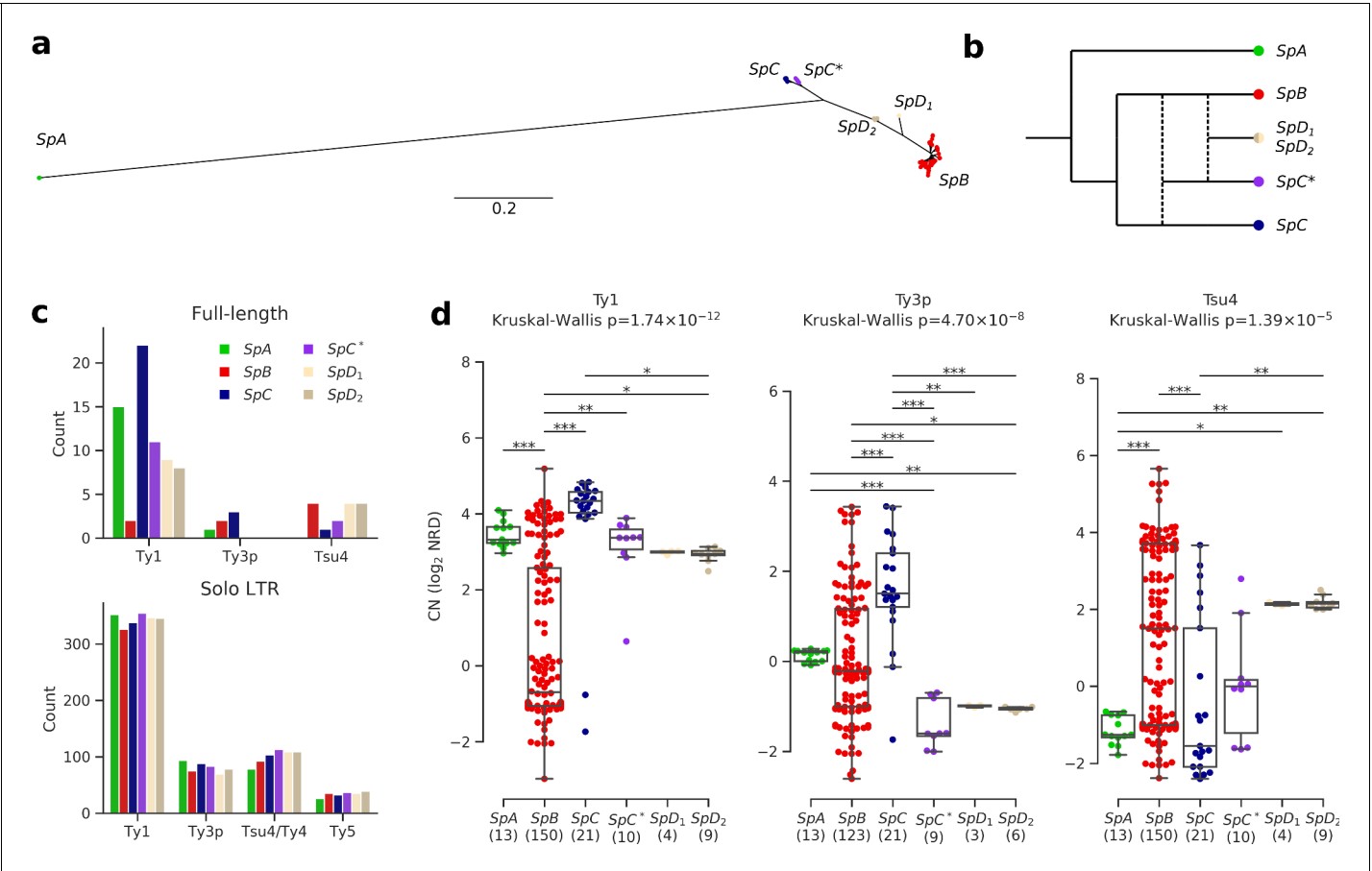

**Figure 2.** Ty elements showed no reactivation in hybrid lineages and contrasted abundances among pure lineages. (a) Bayesian phylogenetic tree of the North American *Saccharomyces paradoxus* lineages based on genome-wide SNPs. (b) Schematic admixture graph showing the evolutionary relationships between pure and hybrid lineages as inferred by *Eberlein et al., 2019*; *Leducq et al., 2016*. Dotted lines represent admixture events. (c) Counts of full-length elements (top) and solo LTRs (bottom) in six whole-genome assemblies based on long reads. No full-length Ty4 or Ty5 were found in any assembly, thus these families were considered extinct. (d) Copy number (CN) variation in active Ty families (i.e. families which have full-length elements) measured as $\log_2$ normalized read depth (NRD) over Ty reference internal sequences for 207 diploid wild strains. Sample size for each lineage (excluding strains for which there was no coverage) is indicated between parentheses. Whiskers span 1.5 times the interquartile range. The p-values of Kruskal-Wallis tests are shown. Stars on bars show the results of Conover post-hoc tests. *: $p<0.05$, **: $p<0.01$, ***: $p<0.001$.

The online version of this article includes the following source data and figure supplement(s) for figure 2:

**Source data 1.** Counts of Ty annotations by type from the six whole-genome assemblies.
**Figure supplement 1.** Counts of LTR retrotransposon annotations in the six whole-genome assemblies.
**Figure supplement 2.** Long terminal repeat (LTR) sequence similarity networks.
**Figure supplement 3.** Depth of coverage on reference internal sequences for Ty1, Ty3p, and Tsu4.

## Results

### *S. paradoxus* lineages exhibit contrasted TE evolutionary dynamics but no reactivation in hybrids

To investigate the TE content and dynamics in North American *S. paradoxus* pure and hybrid lineages, we annotated six highly contiguous whole-genome assemblies previously generated from Oxford Nanopore long reads (PRJNA514804) (*Eberlein et al., 2019*). We used a database comprising reference sequences for *S. cerevisiae* Ty1, Ty2, Ty3, Ty4, and Ty5 families, along with *S. paradoxus* Ty3-related Ty3p (*Carr et al., 2012*) and Ty4-related Tsu4, the latter originating from a horizontal transfer from the *Saccharomyces eubayanus/Saccharomyces uvarum* clade to *S. paradoxus* (*Bergman, 2018*). These annotations revealed counts of full-length elements an order of magnitude lower than solo LTRs (*Figure 2c*, *Figure 2—figure supplement 1*), consistent with previously

reported data on *S. cerevisiae* and *S. paradoxus* (*Carr et al., 2012*; *Goffeau et al., 1996*; *Kim et al., 1998*; *Yue et al., 2017*). To overcome the difficulty of comparing multiple families of short and often degenerated LTR sequences within a phylogenetic framework, we built intra-genome LTR sequence similarity networks. Sequences within the Ty1-Ty2 and Ty3p-Ty3 LTR family pairs were indistinguishable (*Figure 2—figure supplement 2*), in agreement with Ty2 and Ty3 being specific to *S. cerevisiae* (*Carr et al., 2012*; *Naumov et al., 1992*). Consequently, Ty2 and Ty3 LTR annotations were merged with Ty1 and Ty3p, respectively. Although they were clearly distinct, Ty4 and Tsu4 LTRs showed a close relatedness, which justified their combination into the Ty4/Tsu4 family. No Tsu4 LTR was annotated in the *SpA* genome, consistent with this family being specific to North American *S. paradoxus* and absent from Eurasian *S. paradoxus* (*Bergman, 2018*).

Based on the assembly annotations, Ty1 was the most abundant family and exhibited the largest CN variation among lineages, with a striking difference between *SpB* and *SpC* (*Figure 2c*). The generality of CN patterns revealed by genome assemblies was confirmed by normalized short-read depth (NRD) over Ty reference sequences for 207 natural diploid isolates (*Figure 2d*, *Figure 2—figure supplement 3*, *Supplementary file 1a*; PRJNA277692, PRJNA324830, PRJNA479851). The hybrid lineages *SpC\**, *SpD$_1$* and *SpD$_2$* had intermediate full-length Ty1 and Tsu4 CNs compared to their parental lineages, indicating no apparent persistent reactivation or maintenance of Ty elements in these lineages. Notably, Ty3p appeared extinct in all hybrid lineages, with no signal for internal sequences. The most outstanding profile was that of the pure lineage *SpC*, which had significantly higher full-length Ty1 and Ty3p CNs compared to the other lineages. For Tsu4, *SpB* was the lineage with the highest CN. Thus, lineage-wide CNs were consistent with an absence of reactivation in hybrids, the most contrasted profiles being found among pure lineages.

CNs alone may not reflect the actual dynamics of a TE family within a genome. For instance, a recently expanding family may not have had the time to reach the CN of another family (or the same family in a different lineage) that would be more abundant but at equilibrium. We thus explored the evolutionary dynamics of Ty families within each genome. If high CNs were caused by recent family expansions, they should be associated with a signature of low divergence (*Bergman and Bensasson, 2007*). We used the sequence divergence to the most closely related LTR for each LTR annotation to approximate evolutionary divergence since transposition. The importance of low-divergence peaks ($\leq$1%) for Ty1 was significantly higher in *SpC* and *SpA* compared to *SpB* (Fisher's exact test, FDR corrected, *Figure 3a*; see *Figure 3—figure supplement 1* for other divergence thresholds). *SpA* also had a significant underrepresentation of low-divergence LTRs for the composite Ty4/Tsu4 family due to the absence of Tsu4. The height of low-divergence peaks was consistent with CNs of full-length elements (*Figure 2d*). Hybrid lineages *SpC\**, *SpD$_1$* and *SpD$_2$* had Ty1 peaks similar to *SpC*, although less important. They also had no peak for Ty3p, consistent with the absence of full-length elements. Overall, this result indicated that CN variation reflected variation in family dynamics and confirmed that Ty elements exhibited no reactivation in hybrid lineages.

Next, we tested whether orthology between genome-wide Ty insertion loci was consistent with transposition activity. We defined Ty orthogroups from a whole-genome multiple alignment of the six assemblies (*Figure 3b*). Most orthogroups were either private to one genome or conserved across the six (*Figure 3—figure supplement 2*), suggesting that a major fraction of Ty insertion polymorphisms segregated at low frequency in North American lineages. This was consistent with studies in *S. cerevisiae* showing that insertions are mostly fixed or at low frequency (*Bleykasten-Grosshans et al., 2013*; *Carr et al., 2012*). Private orthogroups comprised LTRs that had low divergence, consistent with recent transposition activity (*Figure 3—figure supplement 2*). Private orthogroups were significantly enriched in full-length Ty1 elements (*Figure 3b*), including in hybrid lineages. This indicated that all lineages had active transposition for at least some Ty families. It also suggested that an absence of transposition in hybrid lineages could not explain the absence of reactivation. We called lineage-wide insertion polymorphisms from discordant mappings of the 207 short read libraries to estimate the distribution of insertion allele frequencies. This calling method is noisy and only allows to call non-reference alleles, but should nevertheless provide rough estimates of the insertion allele frequency spectra. Principal component analysis (PCA) on non-reference insertions yielded the expected grouping of strains into well-defined lineages (*Figure 3—figure supplement 3*). The allele frequency distributions showed a similar excess of very-low-frequency alleles for the six lineages (*Figure 3—figure supplement 3*), suggesting that Ty elements were subject to largely uniform evolutionary forces across lineages.

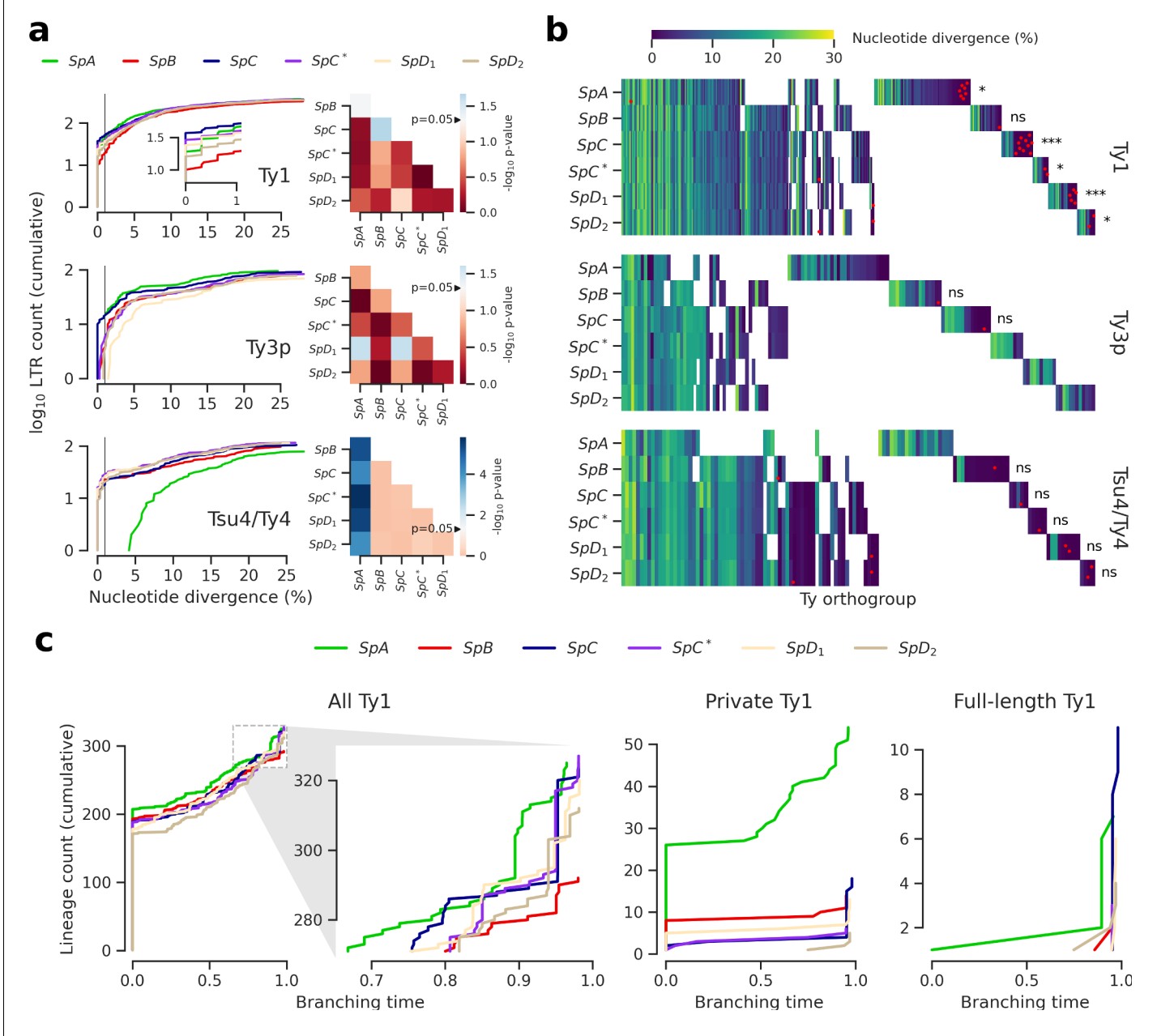

**Figure 3.** Evolutionary dynamics of Ty families in *S. paradoxus*. (a) Distributions of minimum nucleotide divergence between long terminal repeat (LTR) sequences. Heatmaps show FDR-corrected p-values for pairwise Fisher's exact tests between ratios of elements ≤1% and >1% nucleotide divergence. Color maps are centered at the significance threshold of 0.05. Blue cells depict statistically significant tests. See *Figure 3—figure supplement 1* for other divergence thresholds. (b) LTR divergence and conservation levels. Horizontal positions depict Ty orthogroups clustered from conserved (left) to private (right). Color map shows minimum nucleotide divergence. Scattered red dots show LTRs that belong to full-length elements. FDR-corrected p-values of Fisher's exact tests for the ratio of full-length elements in private orthogroups are shown. ns: p≥0.05, *: p<0.05, **: p<0.01, ***: p<0.001. (c) Lineages through time curves for Ty1 ultrametric bayesian phylogenetic trees, using a conserved Ty4 LTR orthogroup as an outgroup. Branching time is scaled such that values of one correspond to the present. From left to right, distributions show all LTRs (with a close-up on recent dynamics), LTRs from private elements and LTRs from full-length elements.

The online version of this article includes the following source data and figure supplement(s) for figure 3:

**Source data 1.** Nucleotide divergence data for LTR sequences from the six whole-genome assemblies.
**Source data 2.** Ty orthogroups defined from the six whole-genome assemblies.
**Source data 3.** Branching times for Ty1 LTR sequences from the six whole-genome assemblies.

*Figure 3 continued on next page*

*Figure 3 continued*

**Figure supplement 1.** Enrichment of low-divergence peaks in distributions of minimum nucleotide divergence between long terminal repeat (LTR) sequences.
**Figure supplement 2.** Ty orthogroups defined from the six genome assemblies.
**Figure supplement 3.** Population structure based on Ty insertions called from discordant short read mappings.

We focused on the most abundant Ty1 to investigate the transposition dynamics in a phylogenetic framework (*Le Rouzic et al., 2013*). Bayesian phylogenetic trees of Ty1 LTRs were converted into ultrametric phylogenies to approximate the timing of branching events using a conserved Ty4 LTR orthogroup as an outgroup. Branching times revealed that lineages had largely similar Ty1 dynamics (*Figure 3c*). Due to the shortness of LTR sequences, most branches collapsed with the outgroup, indicating an expected saturation of the molecular substitution signal. A focus on recent dynamics revealed a clear expansion signature in *SpC* that was shared by *SpC\** and, to a lesser extent, by *SpD₁* and *SpD₂* (*Figure 3c*), which was reflected by the peaks of young private and full-length elements. This suggests that any recent expansion signature in hybrid lineages can be largely explained by inheritance from their *SpC*-related parent. The rapid expansion in *SpC* and *SpC\** can be contrasted with the more gradual expansion in *SpA*, which harbored the second highest Ty1 CN. This contrasted with the assumption that TE abundances can be interpreted as equilibrium CNs under the transposition-selection balance hypothesis (*Charlesworth and Charlesworth, 1983*) and rather suggested a recent transposition rate increase in *SpC*-related lineages.

## Population structure best explains TE content variation, with minor contributions from climatic variation

Our results showed that hybrid lineages exhibited no outstanding Ty CN profiles or sequence dynamics. However, potential hybrid-specific differences in transposition rate might be masked by other factors that can be expected to impact Ty dynamics. To exclude this possibility, we investigated which factors best explain the variation in Ty CN in natural lineages. First, population structure incorporates both host-encoded (*Curcio et al., 2015*) and Ty-encoded (*Czaja et al., 2020*; *Saha et al., 2015*) determinants of Ty transposition rates. Second, environmental factors can influence Ty transposition rate, as demonstrated by temperature (*Paquin and Williamson, 1984*) and nitrogen starvation (*Morillon et al., 2000*) for Ty1 in *S. cerevisiae*. If hybrid lineages experience distinct environmental conditions in nature, this may impact their transposition rates. Finally, the efficiency at which natural selection purges deleterious Ty insertions (which depends on $N_e$) could vary between lineages, ultimately affecting the transposition-selection balance. To help discriminate global from lineage-specific patterns arising from these factors, analyses with all lineages aggregated were complemented by lineage-specific analyses. However, hybrid lineages were represented by relatively few strains ($\leq$10 in each case), which made it hard to reach meaningful conclusions for some analyses. Thus, some of the lineage-specific analyses were focused on the more abundant *SpB* and *SpC* pure lineages.

We first investigated the relative contribution of population structure and environmental factors. We combined genome-wide single nucleotide variants from our population genomic data with climatic (temperature, precipitation, solar radiation, vapor pressure, wind) and water balance data (*Abatzoglou et al., 2018*). We reduced the dimensionality of both datasets by PCA, using coordinates on the major principal components (PCs) to describe population structure (*Figure 4—figure supplement 1*) and climatic variation (*Figure 4—figure supplement 2*, *Figure 4—figure supplement 3*). We then fitted linear models to compare how genotypes-based PCs (gPCs) and environment-based PCs (ePCs) can explain CN variation in Ty1, Ty3p and Tsu4 (*Figure 4a*). Globally, gPCs reached higher levels of statistical significance as linear predictors of CN compared to ePCs. gPC1 both explained a high proportion of variance (52%, 9% and 36% for all lineages, *SpB* and *SpC*, respectively; *Figure 4—figure supplement 1*) and was a highly significant linear predictor of Ty1 CN. Linear models that explained Ty CNs with both gPCs and ePCs had generally lower Akaike information criterion (AIC) values than models that included only gPCs (*Supplementary file 1b*), suggesting that climatic variables have modest although significant contributions in explaining CN variation. Unsupervised clustering of strains along gPCs consistently yielded more homogeneous clusters

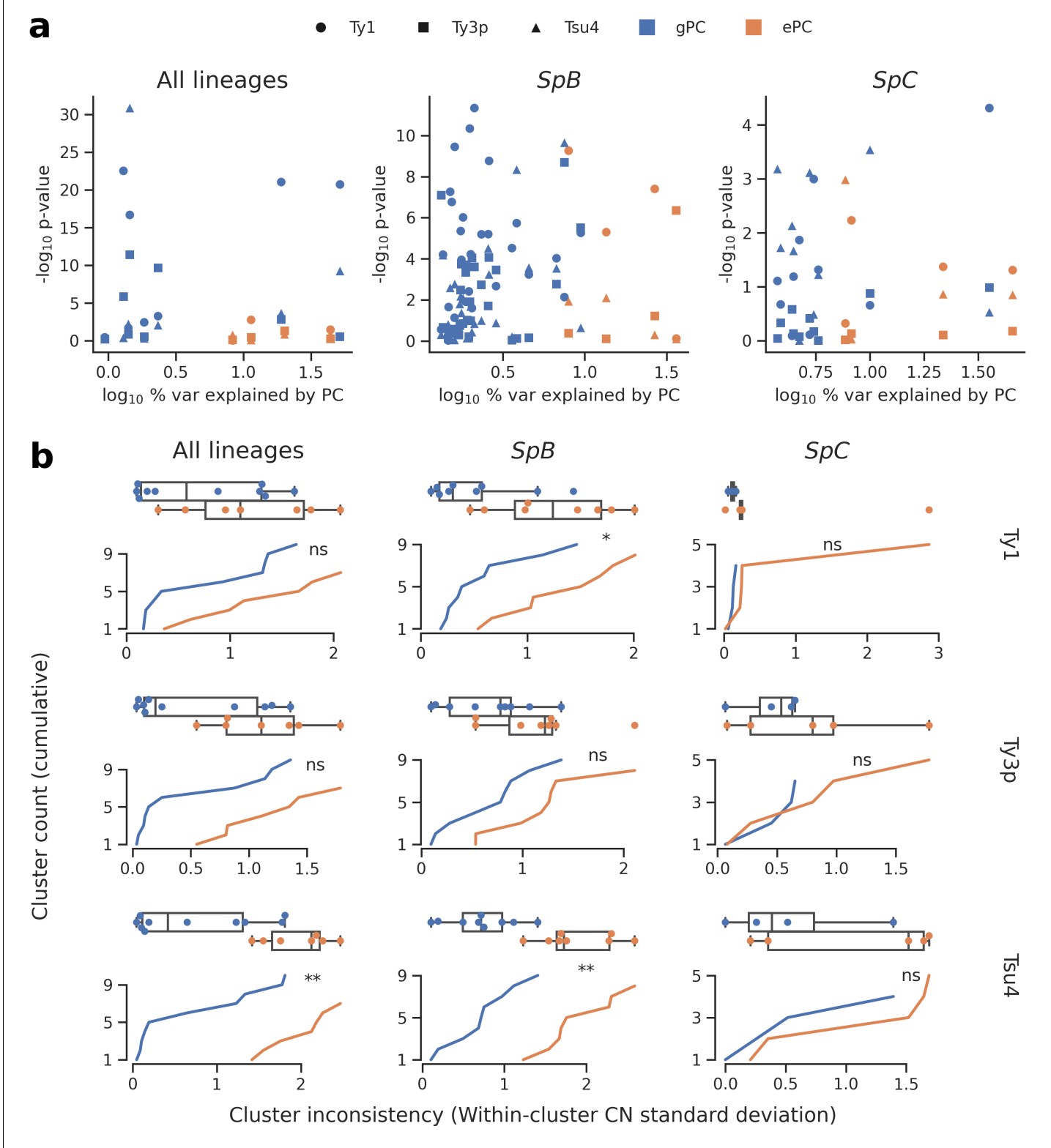

**Figure 4.** Associations between population structure, climatic variables and Ty CNs for all lineages (left), *SpB* only (middle), or *SpC* only (right). (**a**) Linear models fitted on Ty CNs with gPCs and ePCs as independent variables. For each PC, the p-value in the linear model is plotted against the proportion of variance explained in the respective PCA. (**b**) Unsupervised clustering performed on either gPC or ePC coordinates. Cumulative distributions show the standard deviation in Ty CN within each gPC-cluster or ePC-cluster. Distributions are also shown as boxplots, with whiskers

*Figure 4 continued on next page*

*Figure 4 continued*

spanning 1.5 times the interquartile range. FDR-corrected p-values of two-tailed Mann-Whitney U tests between gPC-cluster and ePC-cluster CN distributions are shown. ns: p≥0.05, *: p<0.05, **: p<0.01.

The online version of this article includes the following source data and figure supplement(s) for figure 4:

**Source data 1.** Parameter values for predictors of Ty CNs (gPCs and ePCs) fitted in linear models.
**Source data 2.** gPC-cluster and ePC-cluster membership for wild strains of *S. paradoxus*.
**Figure supplement 1.** Population structure based on genome-wide SNPs.
**Figure supplement 2.** Climatic variation among natural lineages.
**Figure supplement 3.** Loadings of the PCAs on climatic variation data.
**Figure supplement 4.** Strains clustering performed on either gPC or ePC coordinates.

than ePCs in terms of Ty CN (*Figure 4b*, *Figure 4—figure supplement 4*). Overall, these results show that the genetic structure of North American *S. paradoxus* is generally a much better predictor of Ty CNs across lineages than climatic variables. Since locations where hybrid lineages were sampled have climatic profiles similar to pure lineages (*Figure 4—figure supplement 2*), exposure to specific environmental conditions appears unlikely to explain the absence of observable Ty reactivation in hybrid lineages.

## Variation in natural selection efficiency does not explain TE abundance variation in wild lineages

Most TE insertions are expected to have deleterious effects on the individual in which they occur (*Wilke and Adams, 1992*), although some can be advantageous (*Aminetzach et al., 2005*; *Filteau et al., 2015*; *Gresham et al., 2008*; *Van't Hof et al., 2016*). Natural selection acts to purge deleterious insertions, and its efficiency depends on the strength of genetic drift opposing it (which is inversely proportional to $N_e$). Ty reactivation in hybrid lineages might be counterbalanced by higher efficiency of natural selection in these lineages, although there is no a priori reason to expect higher $N_e$ in hybrids. We thus sought to investigate if CN variation across lineages can be explained by variation in $N_e$. Genome-wide nucleotide diversity at fourfold degenerate codon positions ($\pi_S$) is related to long-term $N_e$ and was used to approximate this parameter across diverse taxa (*Lynch and Conery, 2003*). Nucleotide diversity was also used at the intraspecific level in *S. paradoxus* to estimate $N_e$ (*Tsai et al., 2008*). While pure lineages did exhibit variation in $\pi_S$ (*Supplementary file 1c*), there was no obvious negative relationship between Ty CNs and $\pi_S$ (*Figure 5—figure supplement 1*), as would be expected under a transposition-selection equilibrium (assuming negligible variation in mutation rate). For instance, nucleotide diversity was higher in *SpC* than in *SpA*, while *SpC* exhibited the highest Ty1 and Ty3p CN. While these rough $N_e$ estimates should be interpreted with caution, they suggested that natural selection efficiency is likely not the main determinant in CN variation. However, our data indicated that some Ty families experienced recent expansions and thus may not be at equilibrium. Recent hybrid lineages also cannot be meaningfully characterized by long-term $N_e$, further limiting the usefulness of $\pi_S$ as a measure of the efficiency of natural selection.

We can expect that most of the fitness effects associated with TE insertions are caused by alterations of host functions encoded in *cis*. Perhaps the TE insertions with the largest functional alteration potential are those which interrupt ORFs of host genes. Across the six genome assemblies, no Ty insertion was found to interrupt an ORF. Lineage-wide Ty insertion calls revealed four Ty insertions predicted to be within host ORFs, three of which are found in *SpC* strains and one in *SpB*. All the ORF-disrupting insertions were private alleles, and in all cases the strain harboring them had a CN higher than the median of its lineage (*Figure 5—figure supplement 2*). Examination of depth of coverage at these four loci revealed narrow peaks in the corresponding strains that were consistent with target site duplications (TSDs) arising from Ty integration (*Figure 5—figure supplements 3–6*). The absence of ORF-disrupting Ty insertions in hybrid lineages and their scarcity in pure lineages suggest that natural selection is globally very efficient at purging these mutations.

The number of ORF-disrupting Ty insertion alleles in *SpC* (three) compared to *SpB* (one) suggested a slight overrepresentation in *SpC*, which could be explained by a bias towards the accumulation of deleterious insertion alleles in this lineage. To test this hypothesis, we examined the functional annotations of the corresponding genes. There were striking patterns of functional relatedness among genes harboring an ORF-disrupting insertion. Two *SpC* strains had independent Ty1

insertions at different loci within YDL024C (*DIA3*, which encodes a protein of unknown function [*Palecek et al., 2000*]), as evidenced by the corresponding TSDs (*Figure 5—figure supplement 3*, *Figure 5—figure supplement 4*). The two other genes were involved in resistance to arsenic compounds: 16_037 (*SpB*) had a Tsu4 insertion in YPR200C (*ARR2*, which encodes an arsenate reductase [*Mukhopadhyay and Rosen, 1998*], *Figure 5—figure supplement 5*), while LL2011_004 (*SpC*) had a Ty1 insertion disrupting YPR201W (*ARR3*, which encodes an arsenite/H$^+$ antiporter [*Wysocki et al., 1997*], *Figure 5—figure supplement 6*). This functional relatedness could be explained by positive selection; however, this scenario is incompatible with the extremely low frequency at which these insertions segregate. Thus, the most parsimonious explanations for these patterns appears to be either a mutational bias favoring Ty insertions at these loci or fitness effects too small to be visible to natural selection.

We investigated the fitness effect of the ORF-disrupting Ty1 insertion in *ARR3* by measuring the growth of a collection of 123 *S. paradoxus* strains in culture medium supplemented with arsenite (*Figure 5a*, *Figure 5—figure supplement 7*, *Figure 5—figure supplement 8*). Growth in 0.8 mM sodium meta-arsenite (NaAsO$_2$) showed significant variation between lineages (mixed design ANOVA, p=9.54 × 10$^{-39}$), and more total variance was explained by variation between lineages (lineages: 86.58%, strains: 10.20%, residuals: 3.22%). *SpC* and *SpC** lineages exhibited poor growth in 0.8 mM arsenite compared to *SpB* and *SpA* (*Figure 5a*). This phenotype was generalized to the whole lineages, and while LL2011_004 was in the lower part of the *SpC* distribution, it was not significantly different from many *SpC* strains not harboring an insertion (*Figure 5b*, *Figure 5—figure supplement 9*). This indicated that the disruption of a major gene involved in arsenite resistance had no further deleterious effect on the poor growth ability of *SpC*-related strains in presence of arsenite. The absence of growth defect in arsenite for LL2011_004, combined with the extremely low frequencies of all the ORF-disrupting alleles, suggests that these insertions are effectively neutral and are not observed because of an important reduction in purifying selection efficiency.

To gain a more comprehensive view of the strength of selection acting on Ty insertions, we compared the properties of genes in their neighborhood. Ty elements contain many transcription factor binding sites and can affect the expression levels of adjacent genes (*Morillon et al., 2000*; *Coney and Roeder, 1988*; *Todeschini et al., 2005*; *Lesage and Todeschini, 2005*). Dosage-sensitive and functionally central genes should on average be subject to stronger selection against mutations that affect their regulation. For instance, in *S. cerevisiae*, genes that are essential or that have many protein physical interactions were shown to be depleted in eQTLs, i.e. variants that affect expression levels (*Albert et al., 2018*). However, the association of Ty insertions with regulatory effects on adjacent genes is expected to be highly variable, as transcriptional effects are known to depend on the location, orientation and nucleotide sequence of the inserted element (*Coney and Roeder, 1988*; *Lesage and Todeschini, 2005*). Nevertheless, if selection efficiency explains a significant part of Ty CN variation, we reason that when CNs are high (i.e. at low $N_e$), Ty insertions should be more frequently observed near genes which properties reflect stronger dosage and functional constraints, compared to when CNs are low (i.e. at high $N_e$). Focusing on the Ty insertions private to each lineage, we found no significant difference in the proportion of neighboring genes annotated as essential in *S. cerevisiae* (p>0.05, pairwise Fisher's exact tests, FDR correction), although *SpC* did exhibit the highest essential to non-essential ratio (*Figure 5c*). Distributions of physical distance to neighboring genes were similar across lineages, as were distributions of relative orientation (*Figure 5—figure supplement 10*). Additionally, genes neighboring Ty insertions private to *SpC* show no bias towards more protein physical interactions (*Figure 5d*, p>0.05, pairwise two-tailed Mann-Whitney U tests, FDR correction). We fitted the ratio of non-synonymous to synonymous substitutions (ω) to each neighboring gene to estimate the strength of purifying selection on protein-coding sequences. The ω distributions for genes neighboring private Ty insertions exhibited no significant bias (p>0.05, pairwise two-tailed Mann-Whitney U tests, FDR correction). In particular, the distribution in *SpC* exhibited no skew towards lower values, suggesting no association with genes under strong purifying selection (*Figure 5e*). Notably, genes harboring ORF-disrupting insertions show little protein physical interactions and no signature of purifying selection (*Figure 5d–e*).

We next investigated whether lineage-specific selection signatures on genes are associated with Ty insertions. We used the RELAX branch-site codon model (*Wertheim et al., 2015*) to derive estimates of selection intensity (*k*) along individual gene trees. Within the RELAX framework, *k* < 1 indicates relaxed selection and *k* > 1 indicates strengthened (positive or purifying) selection. We define

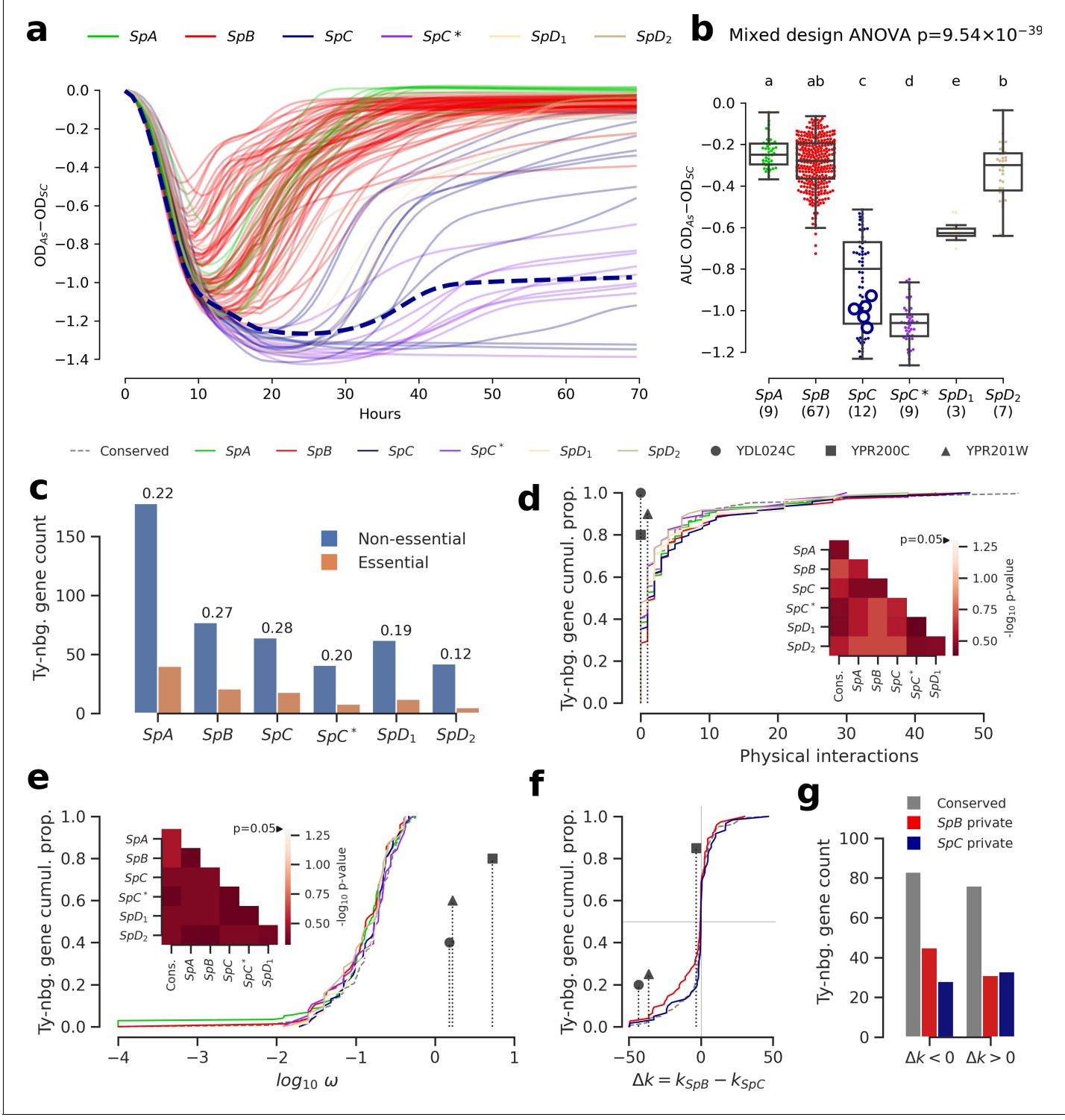

**Figure 5.** Genomic patterns of Ty insertions indicate that natural selection efficiency cannot explain CN variation in natural lineages. (a) Difference between growth curves in 0.8 mM $NaAsO_2$ (As) and control medium (synthetic complete, SC). Curves depict the difference in culture optical density (OD) through time, averaged across replicates and bins of four timepoints (one hour). 123 strains were assayed. *SpC* strain LL2011_004, which harbors a Ty1 insertion in *ARR3*, is represented by a dashed line. (b) Area under the curve (AUC) of the difference between growth curves in As and SC. Replicates of 107 strains that passed the criteria of number of replicates and homoscedasticity are shown. The number of strains per lineage is shown between parentheses. Replicates of the *SpC* strain LL2011_004 are highlighted with large empty circle symbols. Lowercase letters indicate groups that are not significantly different following Tukey's HSD test. (c) Counts of essential and non-essential genes in the immediate vicinity of Ty insertions across the six
*Figure 5 continued on next page*

*Figure 5 continued*

whole-genome assemblies. Essential to non-essential ratios are shown on top of the bars. (d-e) Cumulative distributions of number of protein physical interactions (d) and ω (e) for genes neighboring conserved (dashed line) or private (solid lines) Ty insertions. Genes containing ORF-disrupting insertions are labeled with symbols. Heatmaps show FDR-corrected p-values for pairwise Mann-Whitney U tests between distributions. Color maps are centered at the significance threshold of 0.05. (f) Cumulative distributions of Δk between *SpB* and *SpC*. (g) Counts of neighboring genes with positive or negative Δk values.

The online version of this article includes the following source data and figure supplement(s) for figure 5:

**Source data 1.** Growth data in SC and As for wild *S. paradoxus* strains.
**Source data 2.** Properties of genes neighboring Ty insertions in the six whole-genome assemblies.
**Figure supplement 1.** Synonymous diversity at fourfold degenerate codon positions ($\pi_S$).
**Figure supplement 2.** Ty copy numbers (CNs) of strains harboring ORF-disrupting insertions in relation with lineage-wide distributions of CN.
**Figure supplement 3.** Normalized depth of coverage at locus YDL024C-416.2 predicted to harbor Ty insertions.
**Figure supplement 4.** Normalized depth of coverage at locus YDL024C-416.7 predicted to harbor Ty insertions.
**Figure supplement 5.** Normalized depth of coverage at locus YPR200C predicted to harbor Ty insertions.
**Figure supplement 6.** Normalized depth of coverage at locus YPR201W predicted to harbor Ty insertions.
**Figure supplement 7.** Determination of the optimal $NaAsO_2$ concentration for growth measurements.
**Figure supplement 8.** Growth measurements of the collection of 123 strains in 0.8 mM $NaAsO_2$.
**Figure supplement 9.** Growth variation in $NaAsO_2$ within the *SpC* lineage.
**Figure supplement 10.** Distance and orientation of Ty insertions relative to neighboring genes.
**Figure supplement 11.** Neighbor joining phylogenetic trees of genes with ORF-disrupting Ty insertions.

$\Delta k = k_{SpB} - k_{SpC}$, corresponding to the difference between the average *k* values for clades comprising *SpB* and *SpC* strains, respectively. Overall, the distribution of Δk values for genes neighboring private *SpC* insertions was very similar to that for conserved insertions (*Figure 5f*), while the distribution for private *SpB* insertions was slightly biased toward genes relaxed in *SpB*. For both *SpB* and *SpC*, there were more private insertions adjacent to genes experiencing relaxed selection in the corresponding lineage, although comparisons with conserved insertions were not statistically significant (p>0.05, Fisher's exact tests, *Figure 5g*). This is consistent with efficient purifying selection acting against insertions near genes in both lineages. While genes harboring ORF-disrupting insertions have signatures of stronger selection in *SpC* (*Figure 5e*, *Figure 5—figure supplement 11*), this observation is mitigated by their far-right position in the ω distribution (*Figure 5d*). Overall, these results show that variation in natural selection efficiency is unlikely to explain a large part of the variation in Ty CNs in natural lineages.

## TE content variation in experimental evolution under relaxed selection reveal no systematic effect of hybridization but major effects of individual genotypes

Thus far, our results showed that no reactivation is apparent in natural hybrid lineages, that population structure over environmental variables best explains variation in Ty CNs and that natural selection efficiency is unlikely to explain much variation in Ty genomic landscapes. However, natural selection seems equally efficient at purging deleterious Ty insertions in all lineages, leaving the possibility that strong purifying selection could mask any reactivation in hybrids. Transient reactivation may also be followed by rapid re-establishment of host repression and leave little to no signature in contemporary genomes. Moreover, natural hybrids are somewhat limited in their ability to uncover the interplay between hybridization and transposition, as they are relatively recent introgressed lineages that resulted from admixture between closely related lineages. To systematically explore the effect of hybridization on Ty transposition rates, we performed a large-scale mutation accumulation (MA) experiment on 11 artificial hybrid backgrounds, eight of which were previously published (*Charron et al., 2019*). A MA experiment with yeast consists in subjecting a population to periodic single-cell bottlenecks (typically during hundreds to thousands of mitotic generations) by repeatedly streaking one randomly chosen isolated colony, thus amplifying genetic drift and making natural selection inefficient (*Lynch et al., 2008*; *Figure 6a*). *S. paradoxus* was inferred to reproduce mitotically 99.9% of the time in the wild (*Tsai et al., 2008*), making MA experiments a reasonably realistic approximation of its natural life cycle. Pairs of haploid derivatives of natural *S. paradoxus* and *S. cerevisiae* strains were chosen to span a range of evolutionary divergence (*Figure 6b*),

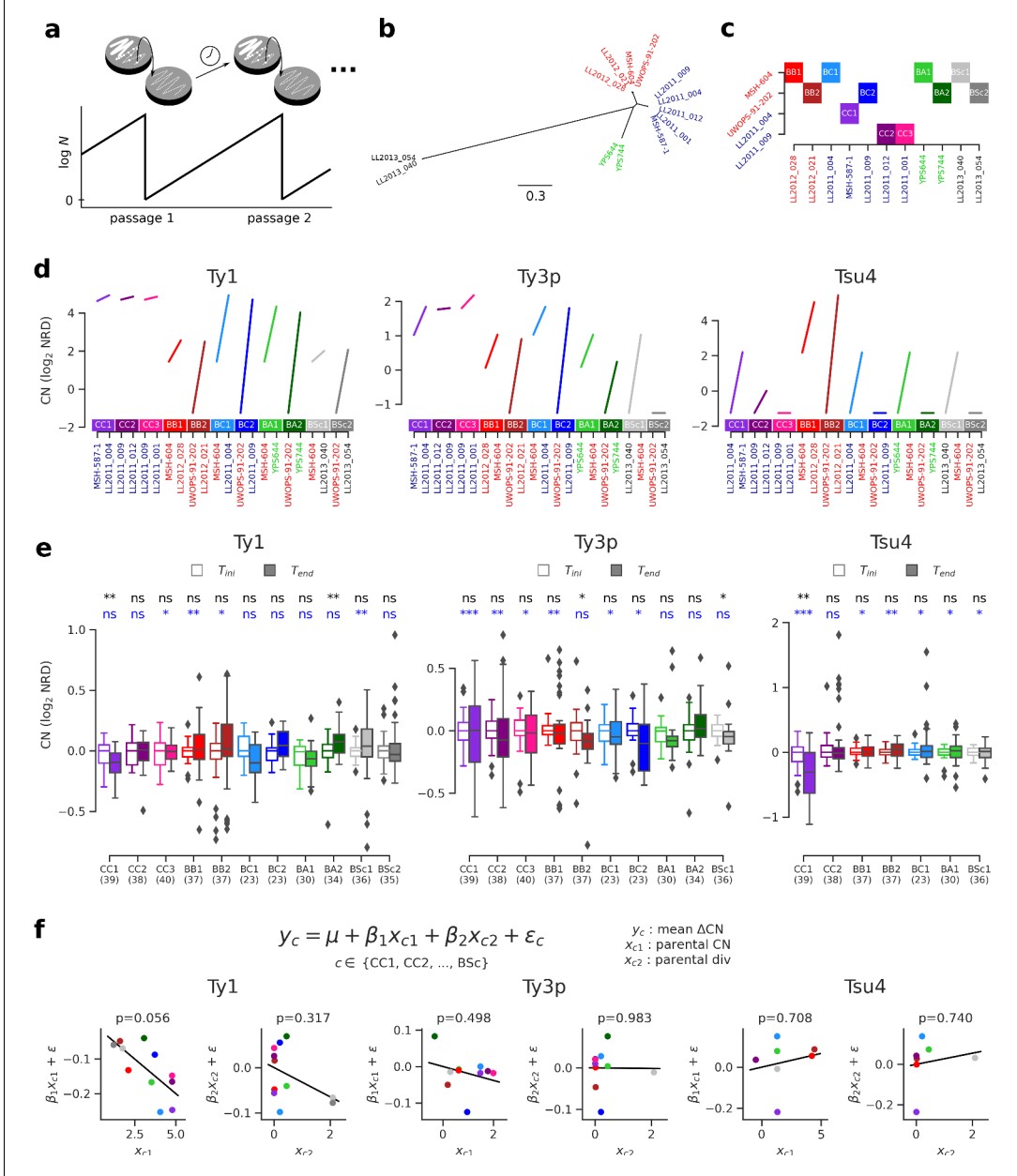

**Figure 6.** Ty copy number (CN) variation in mutation accumulation (MA) lines reveal no effect of parental divergence and major genotype-specific effects. (a) Principle of an MA experiment on a yeast population. *N* corresponds to population size. (b) Bayesian phylogenetic tree of the parental haploid strains used to generate artificial hybrids. (c) Design of the crosses for the MA experiment. (d) Ty CNs in the haploid parental strains of each cross in c. (e) CN at the onset ($T_{ini}$, empty boxes) and at the end ($T_{end}$, filled boxes) of the MA experiment. Whiskers span 1.5 times the interquartile range. FDR-corrected p-values of Wilcoxon signed-rank tests and Brown–Forsythe tests for the equality of variances are shown in black and blue, respectively. ns: $p \geq 0.05$, *: $p < 0.05$, **: $p < 0.01$, ***: $p < 0.001$. (f) Linear models explaining the mean variation in CN ($\Delta$CN = CN$_{Tend}$−CN$_{Tini}$) for a given cross as a function of the joint CN and genetic divergence of its parental strains. The p-value for each term in the models is shown. Scatter plots (CCPR plots) show the partial residuals for each independent variable, along with the fitted regression.

The online version of this article includes the following source data and figure supplement(s) for figure 6:

**Source data 1.** Ty CNs for the haploid parental strains of the MA crosses.

**Source data 2.** Evolutionary divergence between haploid parental strains of the MA crosses.

**Figure supplement 1.** Ty CNs in MA lines with predicted values for the combination of parental CNs.

yielding four categories of intraspecific crosses (from the least to the most divergent): $SpC \times SpC$ (CC), $SpB \times SpB$ (BB), $SpB \times SpC$ (BC) and $SpB \times SpA$ (BA); and one category of interspecific crosses: $SpB \times S.$ *cerevisiae* (BSc; *Figure 6c*, *Supplementary file 1d-f*). The crosses used also spanned a range of variation in parental Ty CNs (*Figure 6d*). For each cross, 48 to 96 MA lines derived from independent matings were propagated for ~770 mitotic divisions.

We selected a random subset of 23 to 40 MA lines per cross and performed whole-genome resequencing on stocks frozen at $T_{ini}$ and $T_{end}$, totaling 372 strains sequenced at each timepoint (PRJNA515073). All these strains were confirmed to be diploid at $T_{ini}$ and $T_{end}$ by quantifying total DNA content using flow cytometry (*Charron et al., 2019*; *Marsit et al., 2020*). Focusing on full-length Ty1, Ty3p, and Tsu4, no global trend emerged from the variation of NRD before and after MA (*Figure 6e*, *Figure 6—figure supplement 1*, *Supplementary file 1f*). Mixed effects linear models fitting cross-specific intercepts and slopes showed no statistically significant support for a global effect of time on CNs. Nevertheless, specific combinations of crosses and Ty families showed significant differences in either central tendency or variance. Often, crosses within a category of parental strain divergence showed contrasting patterns. For example, although CC1, CC2, and CC3 are genetically very similar $SpC \times SpC$ crosses, only CC1 exhibits a significant decrease in Ty1 and Tsu4 CNs. There were also cases of opposite trends within the same category of cross, for instance Ty1 CN in BC and BA crosses. This showed that individual parental genotypes, even if they are evolutionarily closely related, can have very different dynamics, both in sign and magnitude.

We next sought to understand how these dynamics are determined by properties of the different hybrid genotypes tested. We fitted linear models to the variation in CN between $T_{ini}$ and $T_{end}$ ($\Delta$CN), using the sum of CNs of both parental strains of a cross and the evolutionary divergence between them as independent variables. Although these models explain little global variance ($R^2$ = 0.40, 0.08, and 0.06 for Ty1, Ty3p, and Tsu4, respectively), we found that neither independent variable was a significant predictor of $\Delta$CN (*Figure 6f*). Only Tsu4 showed a slightly positive regression slope with respect to evolutionary divergence, and none of them was statistically significant, indicating that divergence between parents of a hybrid has no large influence on its CN dynamics. On the other hand, Ty1 $\Delta$CN had a negative relationship with parental Ty1 CN and was marginally significant (p=0.056), suggesting that the initial CN of a hybrid genotype can dictate the sign and magnitude of its CN change. However, this result should be interpreted with care as some of the crosses shared common parental strains and were thus not entirely independent. This relatedness may in fact explain part of the pattern seen for Ty1. Crosses that shared the *SpB* parental strain UWOPS-91–202 (BB2, BC2, BA2, BSc2) seemed to have systematically higher Ty1 $\Delta$CN values than the corresponding crosses that shared the *SpB* parental strain MSH-604 (BB1, BC1, BA1, BSc1). UWOPS-91–202 appeared to be a naturally Ty-less strain, as no full-length element was found for any family in its genome (*Figure 6d*). Previous reports of such Ty-less strains were shown to be permissive to the accumulation of Ty elements when assayed in the laboratory using a reporter assay (*Czaja et al., 2020*; *Garfinkel et al., 2003*). This observation also supports the hypothesis that genotype-specific characteristics can be major determinants of variation in Ty CN.

## Discussion

The literature on TE reactivation in hybrids demonstrates its relevance to the field of speciation (*Serrato-Capuchina and Matute, 2018*). Reactivation can be a source of postzygotic reproductive isolation by compromising genome stability (*Dion-Côté and Barbash, 2017*), which translates into a spectrum of deleterious hybrid phenotypes spanning sterility, developmental abnormalities and inviability. Many studies investigated the TE reactivation hypothesis at the interspecific level and yielded equivocal evidence. For instance, mixed evidence for TE reactivation was found within the *Drosophila* genus (*Coyne, 1989*; *Hey, 1988*; *Labrador et al., 1999*; *Vela et al., 2014*), where reactivation seems to depend on the species pair involved. On the other hand, three well-established hybrid species that emerged from parallel hybridization between the same species pair in the genus *Helianthus* all exhibit important reactivation of LTR retrotransposons, both at the genomic (*Ungerer et al., 2009*; *Ungerer et al., 2006*) and transcriptomic (*Renaut et al., 2014*) levels. However, a study of hybrid zones between the two parental species revealed that no reactivation is apparent in recent hybrid populations, casting doubt on the idea that hybridization alone triggered reactivation and suggesting the involvement of other factors like abiotic stress (*Kawakami et al., 2011*). Overall, this

data nuances the hypothesis that hybridization could act as a systematic trigger of TE proliferation and reproductive isolation.

In this study, we used population genomic data on North American lineages of the yeast *S. paradoxus* to test whether TEs were reactivated in natural hybrids. It was previously shown that these lineages experienced two hybridization events that resulted in the establishment of admixed lineages that have partial postzygotic reproductive isolation with their respective parental lineages (*Eberlein et al., 2019*; *Leducq et al., 2016*). We found no evidence of TE reactivation in any of these natural hybrid lineages. Our data showed that while some hybrid lineages may have relatively high CNs and exhibit evolutionary dynamics consistent with recent expansions, these signatures were never more pronounced than that of their parental lineages (*SpC* in the case of *SpC\** and *SpC\** in the case of $SpD_1$ and $SpD_2$). Moreover, our analyses suggest that environmental and demographic factors are unlikely to be major factors affecting TE CNs in natural lineages, which therefore likely reflect the underlying transposition rates. Our observations would seem consistent with a role of TE reactivation in speciation, as natural hybrids with high levels of reactivation would likely be unobserved. However, they contrast with long-established hybrid species in *Helianthus*, showing that reactivation and long-term maintenance can be compatible.

We note that the analysis of contemporary genomes is inherently limited in the ability to uncover transient reactivation that would immediately follow hybridization and rapidly faint, perhaps as a result of a re-establishment of TE repression. However, this scenario may be unlikely in yeast, given the absence of many epigenetic and post-transcriptional repression mechanisms (see discussion below) that can yield such a rapid re-establishment in other species. For instance, the re-establishment of piRNA-mediated P element repression in *Drosophila* P-M dysgenic hybrids occurs in single female individuals because paternally inherited piRNA clusters become expressed as they age (*Khurana et al., 2011*). If a hypothetical re-establishment of Ty repression was rather driven by selection for genomic variants, this should result in a gradual improvement of repression with time. Our data provides no support for this hypothesis, as younger hybrid lineages $SpD_1$ and $SpD_2$ do not show more pronounced signatures of recent Ty expansion compared to the older *SpC\** lineage. Additionally, Ty CNs of artificial hybrids at the initial time point of our MA experiment (i.e. early after hybridization) generally show good agreement with the expectation from their parental haploid strains, which is incompatible with an early reactivation of Ty elements.

Many factors are susceptible to determine whether a hybrid genotype exhibits reactivation or not. According to the Bateson-Dobzhansky-Muller (BDM) model, genetic incompatibilities should accumulate as populations diverge (*Orr, 1995*). Additionally, the evolutionary conflict between host genomes and TEs results in an arms race that can accelerate the evolution of the genes involved (*Blumenstiel et al., 2016*; *Van Valen, 1973*). Thus, it can be expected that incompatibilities affecting transposition rates are more likely to be observed in hybrids between highly diverged genotypes compared to closely related ones. Here, we report data from MA experiments on artificial hybrids that probe the relationship between evolutionary divergence and transposition rates. We show that contrary to the expectation from the BDM model, hybrids between more divergent genotypes exhibit no deregulation of TEs. This conclusion is consistent with recent results from Smukowski Heil and colleagues, who showed that no Ty1 reactivation occurs in experimental interspecific hybrids between *S. cerevisiae* and *S. uvarum* (*Smukowski Heil et al., 2020*). Instead of reactivation, many hybrid genotypes exhibited marked decreases in TE CNs after evolution. This trend is especially important for Ty3p, as eight out of ten genotypes exhibit a negative change in median CN. Ty3p is also the least abundant Ty family in natural lineages, as it is found in rarely more than ~5 copies and extinct in all hybrid strains sampled. Such low abundance families are likely more prone to extinction, as stochastic fluctuation in CN would more frequently lead to the loss of all copies. Stochastic loss of Ty copies may come from intra-element LTR recombination or loss of heterozygosity at a locus where a Ty element is present in one parent only, although the mechanistic causes of CN decrease remain to be investigated. This result also seems consistent with the reported Ty CN decrease in mitotically propagated experimental populations of *S. cerevisiae* (*Bast et al., 2019*), suggesting a potential intrinsic bias towards TE loss. However, it should be noted that the latter study used haploid populations in a regime with efficient natural selection (*McDonald et al., 2016*), thus limiting the relevance of comparisons with our results.

Transposition is a complex cellular process, with multiple steps regulated by both TEs and host genomes. Genetic screens in *S. cerevisiae* revealed that genes involved in multiple cellular functions

(chromatin organization, transcription, translation, RNA and protein metabolism) affect Ty1 transposition rate, either positively or negatively (reviewed in *Curcio et al., 2015*). Thus, it is possible that divergent hybrids suffer from misregulation at different steps of the transposition process that have antagonistic effects, such that they cancel out at the level of transposition rates. It is also possible that such antagonistic interactions exist between transposition and excision rates, yielding no net change in CN. Answering these questions will require a thorough investigation of novel Ty insertion loci in the MA lines, which is beyond the scope of this study. Transposition regulation being a polygenic trait with a large mutational target is also consistent with the strong genotype specificity we observed in our artificial hybrids. Some of the contrasted dynamics between crosses that share high genetic similarity might originate from cryptic genetic variation in genes involved in Ty regulation, whose effect on transposition rate would be uncovered in hybrids.

TEs are structurally very diverse and populate the genomes of a vast range of species, and the control mechanisms in response to their mobilization may exhibit a correspondingly large diversity. In many eukaryotes, TEs are repressed epigenetically through mechanisms that include DNA methylation, chromatin modification, and mRNA degradation, processes in which small RNAs play central roles (*Lisch, 2009*; *Slotkin and Martienssen, 2007*). In many fungi, TE repression acts via multiple mechanisms involving RNA interference (RNAi) or cytosine methylation, including repeat-induced point mutation (RIP) (*Gladyshev, 2017*). Notably, introgressed species of the fungal genus *Coccidioides* harbor genomic signatures of efficient RIP (*Neafsey et al., 2010*), suggesting that this mechanism may remain active in the face of hybridization. Unlike other eukaryotes, *S. cerevisiae* and its closely related species lost the genes of the RNAi pathway (*Drinnenberg et al., 2009*) and have no cytosine methylation (*Proffitt et al., 1984*). Instead, *S. cerevisiae* Ty1 has a potent self-repression mechanism, with a strength that depends on the actual Ty1 CN in the host genome (thus termed CN control, CNC) (*Garfinkel et al., 2003*). CNC is a post-transcriptional mechanism that relies on p22, an N-truncated form of the Gag (p49) Ty1-encoded capsid protein which is translated from an alternative transcript of Ty1 (*Saha et al., 2015*). p22 inhibits transposition by interfering with the assembly of the cytoplasmic virus-like particles that are essential to the replication cycle of Ty1 (*Saha et al., 2015*). This unique repression mechanism may explain some of the patterns we observed in the artificial hybrids. In particular, our data suggests a negative relationship between the initial Ty1 CN in hybrid genotypes and the change in Ty1 CN between the onset and the end of the MA experiment. This pattern seems consistent with an efficient action of CNC, although our results would then suggest that CNC efficiency is largely independent of the degree of evolutionary divergence between the subgenomes of a hybrid. This contrasts with the recent finding that interspecific sequence divergence in p22 has a dramatic impact on the action of CNC (*Czaja et al., 2020*), although such impact is likely mitigated by the generally much lower divergence levels in our crosses. Thus, further experimental work is needed to disentangle the mechanistic contributions of Ty1 CN and sequence divergence in the regulation of transposition rates in *Saccharomyces* hybrids.

In conclusion, our work illustrates how the use of natural model systems that are amenable to experimental investigations can bring novel insights on long-standing evolutionary hypotheses. Our results show that natural hybrid lineages of the wild yeast *S. paradoxus* exhibit no TE reactivation and suggest that dynamics of TE content evolution in hybrids depend on deterministic factors that evolutionary divergence between parents can hardly predict.

## Materials and methods

**Key resources table**

| Reagent type (species) or resource | Designation | Source or reference | Identifiers | Additional information |
|---|---|---|---|---|
| Sequence-based reagent | Forward primer to construct the deletion cassette for the *ADE2* gene in *SpC* | *Charron et al., 2019* | CLOP97-F5 | acaattaaggaatcaagaaaccgt gataaaaaattcaagt CAGCTGAAGCTTCGTACGC |
| Sequence-based reagent | Reverse primer to construct the deletion cassette for the *ADE2* gene in *SpC* | *Charron et al., 2019* | CLOP97-F6 | gtaattgttcgctggccaagtata ttaatacatttatata GCATAGGCCACTAGTGGATC |

*Continued on next page*

*Continued*

| Reagent type (species) or resource | Designation | Source or reference | Identifiers | Additional information |
|---|---|---|---|---|
| Strain, strain background (*Saccharomyces paradoxus*) | Haploid parent for MA lines | *Charron et al., 2014* | LL2011_001 *MATa hoΔ::kanMX4* | |
| Strain, strain background (*Saccharomyces paradoxus*) | Haploid parent for MA lines | *This study* | LL2011_001 *MATa hoΔ::kanMX4 ade2Δ::hphNT1* | Haploid yeast strain with *ho* and *ade2* deletions. See Materials and methods section *Mutation accumulation* |
| Strain, strain background (*Saccharomyces paradoxus*) | Haploid parent for MA lines | *Leducq et al., 2016* | LL2011_012 *MATa hoΔ::kanMX4* | |
| Strain, strain background (*Saccharomyces paradoxus*) | Haploid parent for MA lines | This study | LL2011_012 *MATa hoΔ::kanMX4 ade2Δ::hphNT1* | Haploid yeast strain with *ho* and *ade2* deletions. See Materials and methods section *Mutation accumulation* |
| Strain, strain background (*Saccharomyces paradoxus*) | Haploid parent for MA lines | *Charron et al., 2014* | MSH-587–1 *MATa hoΔ::natMX4* | |
| Strain, strain background (*Saccharomyces paradoxus*) | Haploid parent for MA lines | This study | MSH-587–1 *MATa hoΔ::natMX4 ade2Δ::hphNT1* | Haploid yeast strain with *ho* and *ade2* deletions. See Materials and methods section *Mutation accumulation* |
| Strain, strain background (*Saccharomyces paradoxus*) | Haploid parent for MA lines | *Charron et al., 2019* | LL2011_004 *MATα hoΔ::kanMX4 ade2Δ::hphNT1* | |
| Strain, strain background (*Saccharomyces paradoxus*) | Haploid parent for MA lines | *Charron et al., 2019* | LL2011_009 *MATα hoΔ::natMX4 ade2Δ::hphNT1* | |

## General tools

GNU Parallel version 20170922 (*Tange, 2011*) and SnakeMake version 5.5.4 (*Köster and Rahmann, 2012*) were used extensively in the analyses. Unless otherwise stated, Python version 3.7.6 (*Van Rossum and Drake, 2009*) was used for the custom scripts. These scripts were run on Jupyter-Lab version 2.0.1 (*Kluyver et al., 2016*) and made extensive use of the following packages: BioPython version 1.76 (*Cock et al., 2009*), DendroPy version 4.4.0 (*Sukumaran and Holder, 2010*), Matplotlib version 3.2.0 (*Hunter, 2007*), NetworkX version 2.4 (*Hagberg et al., 2008*), NumPy version 1.18.1 (*van der Walt et al., 2011*), pandas version 1.0.2 (*McKinney et al., 2010*), SciPy version 1.4.1 (*Virtanen et al., 2020*), seaborn version 0.10.0 (*Waskom et al., 2020*). Custom scripts can be found in *Supplementary file 2*.

## Protein-coding genes annotation and orthogroup definition

Whole-genome de novo assemblies were previously generated for six North American *S. paradoxus* strains using a combination of long (Oxford Nanopore) and short (Illumina) reads (PRJNA514804). Protein-coding genes were predicted *ab initio* using AUGUSTUS version 3.3.1 with parameters species=saccharomyces genemodel=partial singlestrand=false. Predicted ORF sequences for the six assemblies were combined with ORF annotated from the *S. cerevisiae* strains S288c and YPS128 from *Yue et al., 2017* as references. Protein-coding genes orthogroups were inferred from sequence similarity and synteny using the program SynChro version Jan2015 (*Drillon et al., 2014*) with parameters a=0 DeltaRBH=1. Orthogroups were defined as genes with one-to-one relationships across all pairwise comparisons between the eight genomes. Protein-coding genes systematic names were propagated from S288c to the corresponding orthogroups.

## Annotation of Ty LTR retrotransposons

Ty LTR retrotransposons were annotated from the six whole-genome assemblies using Repeat-Masker version open-4.0.7 (*Smit et al., 2015*) with parameters -s -low -gccalc and the library of reference LTR and internal sequences assembled by *Carr et al., 2012* (containing Ty1, Ty2, Ty3, Ty4, and Ty5 from *S. cerevisiae* and Ty3p from *S. paradoxus*), additionned with Tsu4 from *S. uvarum* (*Bergman, 2018*). RepeatMasker output was defragmented using REannotate version 26.11.2007 (*Pereira, 2008*) with parameters -s 15000 -d 10000, yielding defragmented elements classified as

full-length, solo LTR, truncated or orphan internal sequences. The -f parameter was used to allow defragmentation of Ty1/Ty2 hybrid elements following *Carr et al., 2012*. The defragmented annotations were manually curated to split defragmented elements spanning entire protein-coding genes or forming concatemers.

## Network-based Ty LTR sequence similarity analysis

We analyzed the sequence diversity of LTR annotations within each of our six *S. paradoxus* assemblies using a network approach (*Levy et al., 2017*). The sequences of all LTR annotations within a genome were extracted and all-against-all search was performed between LTR sequences in a genome using BLASTN version 2.7.0 (*Camacho et al., 2009*) with parameters 'word_size 7 dust no'. Networks were built using the Python package Networkx version 2.4 (*Hagberg et al., 2008*) by keeping hits with a bit score higher or equal to the 10th percentile of all bit scores for a given LTR. Visualization of the resulting networks was done using Cytoscape version 3.7.1 (*Shannon et al., 2003*) using the Prefuse Force Directed layout algorithm. Inspection of these networks justified the merging of the following LTR families: Ty1-Ty2, Ty3p-Ty3, and Tsu4-Ty4.

## Estimation of evolutionary divergence since transposition

Evolutionary divergence since transposition was estimated from inter-element LTR sequence identity, assuming that the LTRs of a newly inserted element are identical to those of the source element. From the sequence similarity graphs, the closest relative of each element was retrieved by keeping the hit with the highest bit score. Hits between the two LTRs of full-length elements were excluded, and only one LTR per full-length element was considered. For each element, the evolutionary divergence since transposition was extracted as the percent sequence identity to the closest relative.

## Definition of Ty orthogroups from genome assemblies

Orthology of LTR retrotransposons in the six whole-genome assemblies was assessed as follows. Whole-genome multiple sequence alignment (MSA) with rearrangements was performed using Mugsy version 1r2.3 (*Angiuoli and Salzberg, 2011*) with parameters -d 20000 -nucmeropts '-b 1000 l 15' -fullsearch -refine mugsy. MSA blocks for which at least one genome had no corresponding sequence were filtered out. The remaining alignment blocks were scanned in 10 kb sliding, non-overlapping windows in both directions and windows were trimmed until at least 80% of sites were informative in a MSA window. Ty annotation coordinates were transformed into an MSA coordinates system and clustered using the DBSCAN algorithm implemented in the Python package scikit-learn version 0.22.2 with parameters min_samples=1 eps=500. This one-dimensional clustering approach allowed to define orthologous clusters containing at least one Ty annotation in at least one genome.

We defined a set of confident orthogroups to derive an empirical distribution of physical distance between start and end coordinates of orthologs. Clusters were selected if they contained at most one annotation per genome and if all annotations were on the same strand and belonged to the same family. Pairwise BLASTN searches within each cluster were used to keep elements that produced at least one hit with ≥80% nucleotide identity and an alignment length between 90% and 110% of both query and subject lengths. From the 38 resulting confident orthogroups, we extracted the distribution of differences between annotation start and end coordinates and used the 95th percentile of that distribution (17 bp) as threshold for the following orthogroup definition procedure. Within each cluster, pairs of annotations meeting the following two criteria were considered orthologous: the pair had to include at least one best BLASTN hit, and either start or end coordinates (or both) had to be within the defined threshold of 17 nt. This method allowed to define 713 Ty orthogroups, 20 of which (3%) had >1 element for at least one genome and were discarded.

## Phylogenetic analysis of Ty1 evolution

For each of our six genome assemblies, we extracted the LTR sequence of all Ty1 annotations, keeping only one LTR at random for full-length elements. As an outgroup, a conserved Ty4 LTR orthogroup was added on the basis of minimal variance in start and end coordinates across genomes and minimal difference between the length of the reference Ty4 sequence and the average length of the annotations. MSAs were computed using MUSCLE version 3.8.31 (*Edgar, 2004*) and filtered to maximize the proportion of informative sites while keeping the outgroup, first discarding

sequences with >25% gaps and then removing sites with ≥25% gaps. Bayesian phylogenetic trees were computed using MrBayes version 3.2.5 (*Ronquist et al., 2012*) with model parameters Nst=6 Rates=invgamma. Markov chain Monte Carlo parameters were set to 10 million generations for two runs of four chains each with 25% burn-in fraction and temperature parameter of 0.03 to improve the efficiency of Metropolis coupling. Bayesian phylogenetic trees were converted into ultrametric phylogenetic trees using PATHd8 (*Britton et al., 2007*) and rooted at the Ty4 LTR outgroup. The branching time of each internal node in the ultrametric trees was computed as the distance from the root. Branching events were counted as the number of children nodes for each internal node, as the bayesian phylogenetic reconstructions yielded many polytomies at the base and tips of the trees.

## Construction of a library of reference Ty sequences for *S. paradoxus*

We combined the Ty LTR sequences of all families and full-length sequences of Ty1, Ty3p and Tsu4 from the defragmented annotations of our six *S. paradoxus* assemblies and aligned the sequences of each family and type using MUSCLE. The resulting MSAs were filtered to remove sequences and sites that had ≥70% gaps. Consensus sequences from those MSAs were computed using EMBOSS cons version 6.6 (*Rice et al., 2000*) and the annotation sequence with the lowest BLASTN E value against the resulting consensus was included in the reference library. LTRs of the full-length reference elements were manually trimmed using MEGA version 7.0.26 (*Kumar et al., 2016*). The *S. paradoxus* sequences were complemented with *S. cerevisiae* internal sequences for Ty2, Ty3, Ty4, and Ty5 (*Carr et al., 2012*).

## Ty insertion allele calling and population structure analysis

Illumina paired-end 150 bp or 100 bp libraries for 207 wild *S. paradoxus* diploid strains were downloaded from NCBI (PRJNA277692, PRJNA324830, PRJNA479851) and trimmed using Trimmomatic version 0.36 (*Bolger et al., 2014*) with parameters ILLUMINACLIP:combined_adapters.fa:2:30:10 TRAILING:3 SLIDINGWINDOW:4:15 MINLEN:36 and a custom library of Illumina adapter sequences. Trimmed reads were mapped against the masked version of our *SpB* genome assembly (strain MSH-604, from the RepeatMasker output) using bwa mem version 0.7.16a-r1181 (*Li, 2013*) and sorted using samtools sort version 1.9 (*Li, 2011*). Duplicates were removed using Picard RemoveDuplicates version 2.18.5-SNAPSHOT (*Broad Institute, 2019*) with parameter REMOVE_DUPLICATES=true. RetroSeq version 1.5 (*Keane et al., 2013*) was used to call Ty insertion alleles from discordant pairs of mapped reads. The -discover command was used with parameter -eref and our custom reference Ty sequences library. The -call command was run with parameter -soft. Visual inspection of the resulting calls revealed several instances of multiple, tightly clustered calls of the same family, likely reflecting a single true positive call. To minimize false positives, the calls were transformed into MSA coordinates and clustered using DBSCAN with parameters min_samples=1 eps=500. Resulting clusters were considered true positives if they contained at least one original call of high-quality (flag 8). This strategy provided the best balance between false positive rates and false negative rates in an analysis made with various simulated short-read datasets (data not shown). Insertion allele frequency spectra were computed within each lineage. PCA was performed on Ty alleles using the Python package scikit-learn version 0.22.2.post1.

## Population structure based on genome-wide SNPs

Trimmed reads from the 207 wild strains were mapped against the unmasked version of our *SpB* genome assembly (strain MSH-604) using bwa mem version 0.7.16a-r1181 and sorted using samtools sort. Duplicates were removed using Picard RemoveDuplicates with parameter REMOVE_DUPLICA-TES=true. Variants were called using freebayes version 1.3.1–17-gaa2ace8 (*Garrison and Marth, 2012*). Variants were filtered using bcftools view version 1.9 (*Li, 2011*) with parameter -e 'QUAL<=20' -g ˆhet, along with the tools vcfallelicprimitives, vcfbreakmulti and vcffilter with parameter -f 'TYPE = snp' from vcflib version 1.0.0_rc2 (*Garrison, 2018*). PCA on genome-wide SNPs was performed using the Python package scikit-allel version 1.2.1 (*Miles et al., 2019*), keeping only biallelic SNPs and excluding singletons. From the filtered SNPs of the 207 wild strains, 10000 SNPs were randomly selected and converted to NEXUS format using vcf2phylip version 2.0 (*Ortiz, 2019*). A bayesian phylogenetic tree was computed using MrBayes with model parameters Nst=6

Rates=invgamma. Markov chain Monte Carlo parameters were set to 1 million generations for two runs of four chains each with 25% burn-in fraction and temperature parameter of 0.1.

## Depth of coverage of wild strains on Ty reference sequences

Trimmed reads from the 207 wild strains were mapped against our custom reference Ty sequences library using bwa mem version 0.7.16a-r1181 and sorted using samtools sort version 1.9. Only internal sequences extracted from full-length reference sequences were used. Depth of coverage was extracted using samtools depth. Median depth values of Ty elements were transformed in $\log_2$ and normalized by the $\log_2$ genome-wide depth computed from the mapped bases on the genome-wide mappings previously described. Kruskal-Wallis tests were performed on normalized $\log_2$ depth values for families with full-length elements in natural lineages (Ty1, Ty3p, Tsu4). Conover post-hoc tests were performed to test for pairwise differences between lineages using the Python package scikit-posthocs version 0.6.3 (*Terpilowski, 2019*).

## Climatic data

We used the TerraClimate climatic dataset (*Abatzoglou et al., 2018*) (accessed in October 2019), which incorporates high spatial and temporal resolution climatic interpolation data from primary and derived climatic variables. For the sampling locations of our 207 wild strains, we aggregated the 12 climatic variables over the time period from 1958 to 2015 and computed the mean, standard deviation, minimum and maximum values over time for a given variable variable at a given location. The resulting dataset was normalized into z-scores and summarized by PCA using scikit-learn.

## Association analysis between Ty CNs, population structure and climatic variables

SNP-based population structure PCAs and climatic variable PCAs were combined in linear models as predictors of CNs for the three active families Ty1, Ty3p and Tsu4. To avoid spurious effects arising from log transformation of very sparse coverage, CN values lower than −1.25 were replaced by −1.25 as for MA lines (see below). In each case, PCs explaining a cumulative sum of 80% of the total variance in the dataset were kept and named gPCs (from PCAs on SNP data) or ePCs (from PCAs on climatic variables). Ordinary least squares models were fitted using the Python package statsmodels version 0.10.1 (*Seabold and Perktold, 2010*). Two types of models were fitted: OLS using both gPCs and ePCs as predictors, and OLS using only gPCs as predictors. The quality of the two model types was compared using their AIC values. Strains were clustered according to their coordinates on gPCs using the AffinityPropagation algorithm from scikit-learn with parameters max_iter=10,000 damping=0.8, and subsequently according to their coordinates on ePCs using the KMeans algorithm from scikit-learn with the same number of clusters fitted for gPCs. The inconsistency of each cluster was computed as the standard deviation of CN for each Ty family.

## Nucleotide diversity at synonymous sites

Long-term effective population size for pure lineages was estimated by computing nucleotide diversity at synonymous sites in protein-coding genes. Only SNPs at four-fold degenerate sites in protein-coding genes (third positions of codons CTN, GTN, TCN, CCN, ACN, GCN, CGN, and GGN) were considered. Nucleotide diversity was computed using the mean_pairwise_difference function from scikit-allel, excluding *SpD2* strain WX21 since it is known to be heterozygous (*Eberlein et al., 2019*).

## Determination of the optimal $NaAsO_2$ concentration for growth assay

Three strains were chosen at random in each of the lineages *SpA*, *SpB* and *SpC\** and two *SpC* strains were chosen at random, totalling 12 strains with the *SpC* strain LL2011_004. Precultures were grown in synthetic complete medium (SC; 2% glucose, 0.174% yeast nitrogen base [without amino acids without ammonium sulfate], 0.5% ammonium sulfate, 0.134% complete drop-out) until they reached a 595 nm optical density ($OD_{595}$) of ~0.5–1.0. Precultures were diluted at an $OD_{595}$ of 0.1 ml$^{-1}$ in SC medium supplemented with 0, 0.4, 0.6, 0.8 and 1 mM $NaAsO_2$ in a single 96-well microplate, with border columns and rows filled with sterile SC medium. $OD_{595}$ was monitored during 60 hr at 15 min intervals at 25°C in a Tecan Spark microplate reader (Tecan Group Ltd, Männedorf, Switzerland). $OD_{595}$ values were normalized by subtracting the initial value for each well.

## Growth assay in NaAsO$_2$

From the 207 wild strains used for whole genome sequence analyses, 123 strains had glycerol stocks available. Five replicates of each strain were randomly arranged into four arrays of 14 × 11 positions, totalling 615 replicates with one empty position, with the constraint of having at most three replicates of any given strain within an array. For each array, glycerol stocks of the 123 strains were spotted on YPD plates (1% yeast extract, 2% tryptone, 2% glucose, 2% agar) and incubated at room temperature for three days. Cells from fresh spots were diluted in 200 µl SC medium. OD$_{595}$ was monitored and 500 µl precultures were initiated at 0.08 OD$_{595}$ ml$^{-1}$ and incubated at room temperature for 10 hr. OD$_{595}$ was monitored and precultures were diluted at 0.2 OD595 ml$^{-1}$ in 1 ml SC medium. BRANDplates pureGrade S 384-well microplates (BRAND GMBH + CO KG, Wertheim, Germany) were filled with 40 µl of SC medium (even column numbers) or SC medium supplemented with 1.6 mM NaAsO$_2$ (odd column numbers). 40 µl of cultures at 0.2 OD$_{595}$ ml$^{-1}$ were added to the plates using an Eppendorf epMotion 5075 liquid handling platform (Eppendorf, Hamburg, Germany). Replicates were positioned according to the predefined randomized scheme in side-by-side pairs, yielding adjacent replicates diluted at 0.1 OD$_{595}$ ml$^{-1}$ in SC and SC with 0.8 mM NaAsO$_2$. Border rows and columns of the 384-well microplates were filled with 80 µl sterile medium. Microplate lids were treated with 0.05% Triton X-100 in 20% ethanol for 1 min and air dried (*Brewster, 2003*). OD$_{595}$ was monitored during 70 hr at 15 min intervals at 25°C in a Tecan Spark microplate reader.

OD$_{595}$ values were normalized by subtracting the initial value for each well, and values in SC were subtracted from values in NaAsO$_2$ for each replicate. The area under the curve (AUC) of the resulting curves was obtained by summing all timepoints. Replicates arising from precultures that failed to grow were excluded, and strains which had less than three replicates left were excluded. The standard deviation of replicates was computed for each strain, and strains which fell outside the [−1, 1] z-score range were filtered out to meet the homoscedasticity requirements of analysis of variance (ANOVA). To assess the significance of lineage effects, a mixed effects linear model (with strains as random variable) was fitted in R version 3.4.3 (*R Development Core Team, 2017*) using the lmer function from the package lme4 version 1.1–15 (*Bates et al., 2015*) and an ANOVA table was produced using the package lmerTest version 3.1–2 (*Kuznetsova et al., 2017*). Variance components were estimated using the R package VCA version 1.3.4 (*Schützenmeister and Piepho, 2012*). Post-hoc Tukey's HSD test was performed using scikit-posthocs.

## Properties of genes neighboring Ty insertions

The analysis of properties of neighboring genes considered Ty-centric immediate adjacency relationships with protein-coding genes, meaning that a single gene could be represented more than once if many Ty annotations are immediately adjacent to it. From the whole-genome MSAs, protein-coding genes adjacent to Ty annotations were retrieved on the basis of synteny with the *S. cerevisiae* S288c genome. Genes annotated as essential in *S. cerevisiae* were extracted from YeastMine (accessed in August 2019). Physical interaction data for *S. cerevisiae* S288c were retrieved from the BioGRID Multi-Validated Physical interaction dataset version 3.5.188 (*Oughtred et al., 2019*) and the interactome was built using Networkx.

The strength of purifying selection acting on individual genes was assessed at two levels: at the species-wide level (i.e. including all North American *S. paradoxus* lineages) by fitting a single $\omega$ ratio (non-synonymous to synonymous substitution rates) per gene; and at the lineage level, by evaluating the relative strength of natural selection in the main pure lineages *SpB* and *SpC*. A randomly selected subset of 34 strains from the 207 wild strains were chosen. The VCF matrix from the SNP calling step above was used to derive the DNA sequences of protein-coding genes in these 34 strains. Neighbor-joining gene trees were produced using the nj function from the R package ape version 5.0 (*Paradis and Schliep, 2019*) and R version 3.4.3 (*R Development Core Team, 2017*). $\omega$ ratios were fitted using codeml from the PAML suite version 4.8 (*Yang, 2007*) with parameters seq-type=1 model=0. Selection strength at the lineage level was evaluated using the RELAX framework from the HyPhy software suite version develop-0d598ad (*Wertheim et al., 2015*). The RELAX general descriptive model was fitted using the RELAX-scan script (*Pond, 2019*). From the model output, the fitted value of the $k$ parameter (describing the strength of selection) was extracted for each branch of the gene trees. For *SpB* and *SpC*, an average $k$ value was computed by selecting the clade containing all the strains of the corresponding lineage (from the random subset) and weighting $k$

values by the length of the corresponding branch. The difference in selection intensity between *SpB* and *SpC* was defined as $\Delta k = k_{SpB} - k_{SpC}$. For the three genes with ORF-disrupting Ty insertions, the RELAX framework was fitted on gene trees comprising all 207 wild strains.

## Mutation accumulation

Crosses BB, BC, BA, and BSc were previously published (*Charron et al., 2019*). CC crosses were performed following the same methodology. The *ADE2* gene was deleted in haploid *MATa hoΔ::kanMX4* or *MATa hoΔ::natMX4* derivatives of the *SpC* strains LL2011_001, LL2011_012 and MSH-587–1 (*Charron et al., 2014*) using a *hphNT1* cassette that was amplified from plasmid pFA6a-hphNT1 (*Janke et al., 2004*) using primers CLOP97-F5 and CLOP97-F6 (*Supplementary file 1d,e*). The three strains were paired with the previously published *SpC* strains LL2011_004 *MATα hoΔ::kanMX4* and LL2011_009 *MATα hoΔ::natMX4* to generate three CC crosses by selection on medium containing 100 µg ml$^{-1}$ geneticin and 10 µg ml$^{-1}$ nourseothricin. For each cross, 64 MA lines were initiated by selecting single colonies from independent mating cultures that were spotted on selective medium. Diploid hybrids were propagated mitotically on Petri dishes containing solid YPD medium (1% yeast extract, 2% tryptone, 2% glucose, 2% agar) by streaking for single colonies. Every three days, a single colony was randomly chosen to inoculate a new plate by picking the colony that grew the closest to a predefined mark on the Petri plate. Colonies that had an obvious difference in red pigmentation were excluded to avoid the propagation of mitochondrial petite mutants. MA lines were propagated for 35 passages (approx. 770 mitotic generations), keeping glycerol stocks of all the lines every three passages.

## Illumina library preparation for MA lines

For each cross, 23 to 40 MA lines were randomly chosen among the subset for which diploidy was confirmed at $T_{ini}$ and $T_{end}$ (*Charron et al., 2019*; *Marsit et al., 2020*). The glycerol stocks from passage 1 ($T_{ini}$) and passage 35 ($T_{end}$) were streaked on solid YPD medium. Single colonies were used to inoculate liquid YPD cultures. DNA was extracted from saturated cultures using QIAGEN Blood and Tissue kit with modifications for yeast cell wall digestion. DNA was treated with RNase A and purified on Axygen AxyPrep Mag PCR Clean-up SPRI beads. Genomic DNA libraries were prepared using the Illumina Nextera Tagmentase and Nextera index kits following a modified version of the protocol described in *Baym et al., 2015*. The quality of randomly selected libraries was controlled using an Agilent BioAnalyzer 2100 electrophoresis system. Sequencing was performed at the CHU Sainte-Justine and Génome Québec Integrated Centre for Pediatric Clinical Genomics (Montreal, Canada) on an Illumina NovaSeq 6000 sequencer with SP flow cells.

## Depth of coverage of MA lines on Ty reference sequences

Illumina paired-end 150 bp libraries for 372 MA lines and their 13 parental haploid strains (deposited at NCBI Sequence Read Archive under PRJNA515073) were trimmed using Trimmomatic version 0.33 with parameters ILLUMINACLIP:nextera.fa:6:20:10 MINLEN:40 and a library of Illumina Nextera adapter sequences. Trimmed reads were mapped against our *SpB* genome assembly (strain MSH-604) using bwa mem version 0.7.17 and sorted using samtools sort version 1.8. Duplicates were removed using Picard RemoveDuplicates version 2.18.29-SNAPSHOT with parameter REMOVE_DUPLICATES=true. We extracted the list of *S. cerevisiae* genes with paralogs from YeastMine (accessed in December 2018) (*Balakrishnan et al., 2012*) and randomly selected 50 genes that did not figure in this list. The protein sequences of these genes were used as TBLASTN queries against our *SpB* genome assembly and the best hit loci were extracted. Depth of coverage over these loci was extracted using samtools depth version 1.9 and depth values were transformed in log$_2$. Depth values were transformed in log$_2$ and normalized by the log$_2$ genome-wide depth computed from the mapped bases on the genome-wide mappings.

 The trimmed reads of the 372 MA lines and their 13 parental haploid strains were mapped against our custom reference Ty sequences library using bwa mem version 0.7.16a-r1181 and sorted using samtools sort version 1.9. Only internal sequences extracted from full-length reference sequences were used. Median depth values of Ty elements were transformed in log$_2$ and normalized by the log$_2$ genome-wide computed from the mapped bases on the genome-wide mappings. Normalized log$_2$ depth values for a given cross and Ty family were discarded if the median was lower than the

10th percentile of the normalized $\log_2$ depth values for non-paralogous genes in the corresponding cross, thus considering the family absent from the initial MA genotypes. Similarly, the 10th percentile of normalized $\log_2$ depth values for non-paralogous genes in all crosses aggregated ($-1.25$) was used as a lower threshold for depth values of the parental haploid strains, replacing lower values by $-1.25$.

## Phylogenetic analysis of the MA lines haploid parents

From the mappings of the MA lines parental haploid strains, variants were called using freebayes. Variants were filtered using bcftools view with parameter -e 'QUAL <= 20 g 'het', along with the tools vcfallelicprimitives, vcfbreakmulti and vcffilter with parameter -f 'TYPE = snp' from vcflib. From the filtered SNPs, 10000 SNPs were randomly selected and converted to NEXUS format using vcf2phylip. A bayesian phylogenetic tree was computed using MrBayes with model parameters Nst=6 Rates=invgamma Ploidy=haploid. Markov chain Monte Carlo parameters were set to 1 million generations for two runs of four chains each with 25% burn-in fraction and temperature parameter of 0.1. From the resulting consensus tree, the evolutionary divergence between parental strains was measured as the sum of branch lengths connecting each pair.

## Association analysis between Ty ΔCN, evolutionary divergence between parents and initial Ty CN

For each cross, the average ΔCN was analysed in a linear model incorporating parental evolutionary divergence and initial Ty CN of the hybrid genotypes as predictors. For each cross, the Ty CNs of parental haploid strains were summed and used as a measure of the initial Ty CN in all MA lines of the cross. An OLS model was fitted using statsmodels.

# Acknowledgements

We thank Hélène Martin and Anna Fijarczyk for contributing to sequencing data analyses and Philippe Després, Anne-Marie Dion-Côté, Marika Drouin, Alexandre Dubé, Clara Dubernard and Anna Fijarczyk for reading and providing useful comments on draft versions of this manuscript.

# Additional information

## Competing interests

Christian R Landry: Reviewing editor, *eLife*. The other authors declare that no competing interests exist.

## Funding

| Funder | Grant reference number | Author |
|---|---|---|
| Natural Sciences and Engineering Research Council of Canada | RGPIN-2015-03755 | Christian R Landry |
| Natural Sciences and Engineering Research Council of Canada | NSERC Alexander Graham-Bell doctoral scholarship | Mathieu Hénault Guillaume Charron |
| Fonds de recherche du Québec – Nature et technologies | FRQNT doctoral scholarship | Mathieu Hénault Guillaume Charron |
| Fonds de Recherche du Québec - Santé | FRQS postdoctoral scholarship | Souhir Marsit |
| Canada Research Chairs | Canada Research Chair in Evolutionary Cell and Systems Biology | Christian R Landry |

The funders had no role in study design, data collection and interpretation, or the decision to submit the work for publication.

## Author contributions

Mathieu Hénault, Conceptualization, Data curation, Software, Formal analysis, Funding acquisition, Investigation, Visualization, Methodology, Writing - original draft, Writing - review and editing; Souhir Marsit, Guillaume Charron, Funding acquisition, Investigation, Methodology, Writing - review and editing; Christian R Landry, Conceptualization, Supervision, Funding acquisition, Methodology, Writing - review and editing

## Author ORCIDs

Mathieu Hénault (iD) https://orcid.org/0000-0003-0760-7545
Christian R Landry (iD) http://orcid.org/0000-0003-3028-6866

## Decision letter and Author response

Decision letter https://doi.org/10.7554/eLife.60474.sa1
Author response https://doi.org/10.7554/eLife.60474.sa2

# Additional files

## Supplementary files

• Supplementary file 1. Supplementary tables. (a) Description of the wild strains used in this study. CNs of full-length Ty1, Ty3p and Tsu4 (expressed as $\log_2$ NRD) are indicated for individual strains. (b) AIC values for linear models testing the association between population structure and environmental variation. (c) Nucleotide diversity at four-fold degenerate sites in protein-coding genes ($\pi_S$). (d) Primers used for *ADE2* deletion in the parental *SpC* strains generated in this study. Primers were described in *Charron et al., 2019*. (e) Parental strains used in the MA experiment made in this study. (f) Description of the MA lines used in this study. CNs of full-length Ty1, Ty3p and Tsu4 (expressed as $\log_2$ NRD) are indicated for individual strains.

• Supplementary file 2. Custom scripts used for data analyses.

• Transparent reporting form

## Data availability

Illumina short read sequencing data of the MA lines is available at NCBI under accession PRJNA515073. Genome assemblies and Nanopore long read sequencing data of wild isloates are available at NCBI under accession PRJNA514804. Illumina short read sequencing data of wild isolates is available at NCBI under accessions PRJNA277692, PRJNA324830 and PRJNA479851.

The following dataset was generated:

| Author(s) | Year | Dataset title | Dataset URL | Database and Identifier |
|---|---|---|---|---|
| Marsit S, Henault M | 2020 | Ploidy instability in experimental hybrid populations | https://www.ncbi.nlm.nih.gov/bioproject/?term=PRJNA515073 | NCBI BioProject, PRJNA515073 |

The following previously published datasets were used:

| Author(s) | Year | Dataset title | Dataset URL | Database and Identifier |
|---|---|---|---|---|
| Leducq JB | 2016 | Speciation driven by hybridization and chromosomal plasticity in a wild yeast | https://www.ncbi.nlm.nih.gov/bioproject/?term=PRJNA277692 | NCBI BioProject, PRJNA27769 |
| Anderson JB | 2017 | Saccharomyces paradoxus Raw sequence reads | https://www.ncbi.nlm.nih.gov/bioproject/?term=PRJNA324830 | NCBI BioProject, PRJNA324830 |
| Eberlein C, Henault M | 2019 | Hybridization is a recurrent evolutionary stimulus in wild yeast speciation | https://www.ncbi.nlm.nih.gov/bioproject/?term=PRJNA479851 | NCBI BioProject, PRJNA479851 |

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
