## [Decision Letter]

**Acceptance summary:**

We appreciate that your study furthers our insight into the factors that control transposon activity. It seems that in contrast to what has been suggested previously, hybrids do not always show an elevated transposon activity, and that the genetic background (and complex interactions between alleles/mutations) may be a more important influence on transposon activity.

**Decision letter after peer review:**

Thank you for submitting your article "The effect of hybridization on transposable element accumulation in an undomesticated fungal species" for consideration by *eLife*. Your article has been reviewed by three peer reviewers, one of whom is a member of our Board of Reviewing Editors, and the evaluation has been overseen by Patricia Wittkopp as the Senior Editor. The following individual involved in review of your submission has agreed to reveal their identity: Douda Bensasson (Reviewer #3).

The reviewers have discussed the reviews with one another and the Reviewing Editor has drafted this decision to help you prepare a revised submission.

All reviewers appreciate the relevance and scientific rigor of your study. Still, all reviewers also agree that, despite the large body of work presented here, it may not be possible to conclude that hybridization never induces Ty activity (or reduces repression of Ty activity), as this may depend on the exact nature (evolutionary distance, specific alleleles…) of the parental genomes. While you are very careful with interpreting the results, we would ask you to more explicitly discuss these possible alternative scenarios.

Two reviewers also suggested using established reporter assays to directly measure Ty activity in hybrids. In principle, these are very easy experiments that would, in our opinion, provide a fantastic way to consolidate the conclusions. However, we want to leave it up to you to decide whether you want to include this in the current manuscript.

Reviewer #1:

In this paper, Hénault and colleagues investigate the hypothesis that transposons (Ty elements) may become more active in hybrids. They first explore the genomes of natural hybrid *S. paradoxus* lineages and do not find evidence for elevated Ty activity in hybrid lineages, although some strains exhibit signs of higher Ty activity than others. Next, a mutation-accumulation experiment where Ty activity is measured in 6 lineages of hybrids with varying degrees of heterozygosity further confirms these findings.

Overall, this is a well-written and interesting study that helps to gain further insight into the factors that control transposon activity. The results suggest that, in contrast to what has been suggested previously, at least not all hybrids show an elevated transposon activity, and that the genetic background (and complex interactions between alleles/mutations) may be a more important factor. That said, I am a bit puzzled by the way the problem was approached – I appreciate that forensics of natural genomes is perhaps the best way to investigate a natural genetic phenomenon, but it obviously also has shortcomings. The lab experiments are a great addition, but there I cannot help but wondering whether using reporter systems for Ty activity (e.g. the typical systems with fluorescent reporters that have an intron) would have not been a more efficient way to measure Ty activity in thousands of different (artificial and natural) hybrids? Perhaps substance for a follow-up study…

Specific questions and concerns:

I wonder to what extent the genome analysis (i.e. the first part of the paper) depends on how long ago the hybridization events occurred. It seems plausible that if Ty elements become more active, this effect is the biggest immediately after the hybridization, and disappears over time as the genome re-establishes suppression of Ty activity. Would this influence the findings? It seems useful to discuss this in the text.

I suspect that using third-position codon diversity is a rather poor estimate of an effective population size, especially over longer periods of time? If this is the case, perhaps the results using this analysis should be interpreted with even more caution?

Is it necessarily true that the overrepresentation of ORF-disrupting Ty insertions in the *SpC* lineage points at a bias towards more deleterious insertions? Are the numbers strong enough to exclude noise? Or could selection be lower/more recent here?

Is it necessarily true that we expect a negative correlation between Ty insertions near essential genes and a lower purifying selection? Could noise play a role? And, how strong is the effect that a mutation (or more accurately, a Ty insertion) near an essential gene is on average more deleterious? Do we know how far that effect stretches from the gene, and was this taken into account in the analysis?

Reviewer #2:

I think the topic is very interesting, the work is well done, and the manuscript is well written (maybe can be shorten). Overall, the message is that the hybridisation does not trigger TE expansion in yeast and is supported by population genomics and experimental evolution data. I have two main points:

1) Did the authors were too conservative with their hybrid design and crossed populations that were too similar? I guess their rational is to mimic what they isolated in nature, but maybe if they look at the *S. cerevisiae* x *S. paradoxus* crosses that they published before, this might change.

2) I think the natural strains described in the first part of the paper and used for the crossing, would benefit for characterising TE activity, e.g. looking expression or better using existing reporters that enable to calculate rates of retrotransposition and excision. I think this can have major impact in the results obtained from the MALs and is a more direct read out of the hybridisation effect.

Reviewer #3:

This work uses wild yeast as a model and thus integrates many approaches to test competing theories about transposable element (TE) contributions to early speciation. It shows that natural hybrids between *S. paradoxus* subspecies show TE copy numbers (CNs) that are intermediate between those of their parents. Population genetic structure explains most of the natural variation among strains in TE levels though there could be a small effect of climate on Ty3p CNs. There is no evidence that TEs are at higher levels because of random drift. Mutation accumulation experiments with good replication similarly show no increase in transposition in hybrids compared with non-hybrid controls. Past work showed that the model yeast lineages used here are partially reproductively isolated. This work shows that their hybridization does not lead to increased TE reactivation and is not driving their speciation through Bateson-Dobzhanski-Muller interactions.

The experiments and analyses are convincing, and I have no substantive concerns. Furthermore, this number of integrated analyses is needed and would only be possible in few eukaryotes. The authors present a thorough discussion that interprets findings in the context of the model system and literature for all eukaryotes, from molecular biology to evolutionary genetics.

---

## [Author Response]

Reviewer #1:[…] That said, I am a bit puzzled by the way the problem was approached – I appreciate that forensics of natural genomes is perhaps the best way to investigate a natural genetic phenomenon, but it obviously also has shortcomings. The lab experiments are a great addition, but there I cannot help but wondering whether using reporter systems for Ty activity (e.g. the typical systems with fluorescent reporters that have an intron) would have not been a more efficient way to measure Ty activity in thousands of different (artificial and natural) hybrids? Perhaps substance for a follow-up study…

We thank the reviewer for the positive feedback and for the constructive suggestion of using retrotransposition reporter assays as an orthogonal approach. To avoid repeating our response, we invite the reviewer to see the response to reviewer 2’s second major comment, which expressed basically the same suggestion.

Specific questions and concerns:I wonder to what extent the genome analysis (i.e. the first part of the paper) depends on how long ago the hybridization events occurred. It seems plausible that if Ty elements become more active, this effect is the biggest immediately after the hybridization, and disappears over time as the genome re-establishes suppression of Ty activity. Would this influence the findings? It seems useful to discuss this in the text.

We agree with the reviewer that the timing of hybridization may have an impact on the ability to observe Ty reactivation. It is entirely possible that a reactivation immediately following hybridization may have been rapidly suppressed by re-establishment of host repressive mechanisms, and that most resulting insertions were further purged by selection or lost by drift. Thus, this approach has limited ability to detect past, transient episodes of reactivation. We are aware that this is an important drawback of the population genomic approach, as highlighted by the reviewer. Thus, we modified the text to emphasize that our ability to detect reactivation from contemporary genomes is limited to persistent reactivation or to the maintenance of the accumulated TEs.

When first presenting the CN profiles from wild lineages, we added:

“The hybrid lineages *SpC**, *SpD_1_* and *SpD_2_* had intermediate full-length Ty1 and Tsu4 CNs compared to their parental lineages, indicating no apparent persistent reactivation or maintenance of Ty elements in these lineages.”

When transitioning from the population genomics section to the experimental evolution section, we emphasized the difficulty to detect transient reactivation from contemporary genome analysis by adding this sentence: “Transient reactivation may also be followed by rapid re-establishment of host repression and leave little to no signature in contemporary genomes.”

Finally, we added a paragraph in our Discussion to mention this limitation and discuss the likelihood of transient reactivation scenarios in the light of our results and the known repression mechanisms in yeast:

“We note that the analysis of contemporary genomes is inherently limited in the ability to uncover transient reactivation that would immediately follow hybridization and rapidly faint, perhaps as a result of a re-establishment of TE repression. […] Additionally, Ty CNs of artificial hybrids at the initial time point of our MA experiment (i.e. early after hybridization) generally show good agreement with the expectation from their parental haploid strains, which is incompatible with an early reactivation of Ty elements.”

I suspect that using third-position codon diversity is a rather poor estimate of an effective population size, especially over longer periods of time? If this is the case, perhaps the results using this analysis should be interpreted with even more caution?

We agree with the reviewer that non-synonymous diversity at codon third positions only provides very rough estimates of *N*_e_. While we tried to state this in the text, we understand that readers may easily overestimate the level of confidence we put in this result. We thus changed the main text to emphasize the suggestive nature of this result. We also cited Tsai et al., 2008 as an example of the use of nucleotide diversity in population genomic data to estimate population genetic parameters.

“Genome-wide nucleotide diversity at four-fold degenerate codon positions (π_S_) is related to long-term *N*_e_ and was used to approximate this parameter across diverse taxa (Lynch and Conery, 2003). […] While these rough *N*_e_ estimates should be interpreted with caution, this result suggested that natural selection efficiency is likely not the main determinant in CN variation.”

Is it necessarily true that the overrepresentation of ORF-disrupting Ty insertions in the SpC lineage points at a bias towards more deleterious insertions? Are the numbers strong enough to exclude noise? Or could selection be lower/more recent here?

Although many of our results tend to suggest that the *SpC* lineage could be subject to relaxed selection compared to others, we want to emphasize that these results constitute mixed evidence and we do not believe that the higher count of ORF-disrupting insertions in *SpC* compared to *SpB* (3 vs 1) is strong evidence in support for this hypothesis. We cannot exclude that it could be due to chance or sampling bias. The results we obtained in the search for a phenotypic defect associated with the private Ty1 insertion in *ARR3* indicate that this polymorphism is probably neutral, since the single strain bearing it has no significant growth defect in arsenite compared to the rest of the *SpC* lineage. We conclude from this result that relaxed selection cannot explain the occurrence of this rare polymorphism and speculate that this interpretation could apply to other ORF-disrupting insertions. We thank the reviewer for highlighting that this was not easily understood from our previous formulation, and we modified the text to make it clear that we do not claim any selective interpretation about the ORF-disrupting insertions.

We edited the beginning of the first paragraph of this section:

“The number of ORF-disrupting Ty insertion alleles in *SpC* (three) compared to *SpB* (one) suggested a slight overrepresentation in *SpC*, which could be explained by a bias towards more deleterious insertion alleles in this lineage. To test this hypothesis, we examined the functional annotations of the corresponding genes.”

To strengthen our estimation of the phenotypic effect of the private Ty1 insertion in *ARR3*, we replicated the phenotypic measurement in arsenite at a larger scale. We modified the fourth paragraph of this section as follows:

“We investigated the fitness effect of the ORF-disrupting Ty1 insertion in ARR3 by measuring the growth of a collection of 123 *S. paradoxus* strains in culture medium supplemented with arsenite (Figure 5A, Figure 5—figure supplement 7, Figure 5—figure supplement 8). […] The absence of growth defect in arsenite for LL2011_004, combined with the extremely low frequencies of all the ORF-disrupting alleles, suggests that these insertions are effectively neutral and are not observed because of an important reduction in purifying selection efficiency.”

Figure 5 was modified. The previous Figure 5A-B was modified and added as a preliminary result as Figure 5—figure supplement 7. Replicates for all phenotypic measurements in NaAsO_2_ were added as Figure 5—figure supplement 8. Statistical analysis of phenotypic variation within the *SpC* lineage was added as Figure 5—figure supplement 9:

We added this section to the Materials and methods:

“Growth assay in NaAsO_2_

From the 208 wild strains used for whole genome sequence analyses, 123 strains had glycerol stocks available. [...] Post-hoc Tukey’s HSD test was performed using scikit-posthocs.”

Is it necessarily true that we expect a negative correlation between Ty insertions near essential genes and a lower purifying selection? Could noise play a role? And, how strong is the effect that a mutation (or more accurately, a Ty insertion) near an essential gene is on average more deleterious? Do we know how far that effect stretches from the gene, and was this taken into account in the analysis?

We agree with the reviewer that our explanation of the rationale we put forward to use adjacency to essential genes as a measure of selection efficiency lacks precision and nuance. Although it is noisy and surely has limited sensibility, this approach is expected to capture large variation in natural selection efficiency acting on the regulation of essential gene expression. Thus, we moderate our claims on the interpretability of these patterns and mention the expected sources of noise, which are indeed important.

Essential genes are known to be dosage-sensitive. Since by definition loss of function is lethal for these genes, important reduction in expression levels is expected to be deleterious for this class of genes. Additionally, they are overrepresented among genes that tolerate little overexpression (Makanae et al., 2013). They also have less genetic variation in mRNA levels (eQTLs) than most genes (Albert et al., 2018), showing that their expression levels are likely constrained by stabilizing selection. If natural selection is less efficient in a given lineage, we expect mutations that affect gene expression levels to be more frequently observed for this subset of genes compared to other lineages.

It is relatively well established that Ty elements can affect neighboring gene expression levels (Morillon, Springer and Lesage, 2000; Coney and Roeder, 1988; Todeschini et al. 2005; Lesage and Todeschini, 2005), either by increasing or decreasing them. Not all Ty insertions affect the expression of neighboring genes, as it depends on characteristics of the individual Ty insertions, notably position and orientation relative to the gene’s regulatory sequences. Although the study of many Ty1 insertions in *S. cerevisiae* allowed to define rules regarding the effect of position and orientation on neighboring gene expression (Lesage and Todeschini, 2005), to our knowledge, the generality of these rules at the genome scale were not assessed. Moreover, it was shown that single nucleotide substitutions can have dramatic consequences on the repressing or activating ability of a Ty element (Coney and Roeder, 1988), adding complexity to the prediction of gene expression effects caused by Ty insertions.

To address these aspects, we moved the background elements on the expression effects on adjacent genes from the second paragraph to the fifth paragraph of the section, and complemented it as follows:

“To gain a more comprehensive view of the strength of selection acting on Ty insertions, we compared the properties of genes in their neighborhood. […] Nevertheless, if selection efficiency explains a significant part of Ty CN variation, we reason that when CNs are high, Ty insertions should be more frequently observed near genes which properties reflect stronger dosage and functional constraints, compared to when CNs are low.”

In an attempt to add robustness to our analysis, we analyzed the distributions of number of protein-protein interactions (i.e. degree in the interactome) for genes neighboring Ty insertions, which corroborates our results based on gene essentiality, ω and *k*. We added the following sentences to the fifth paragraph of this section:

“Additionally, genes neighboring Ty insertions private to *SpC* show no bias towards higher number of physical interactions (Figure 5D, p>0.05, pairwise two-tailed Mann-Whitney U tests, FDR correction).”

We modified Figure 5 by adding a panel showing the distributions of counts of physical interactions, as detailed in the previous comment about ORF-disrupting Ty insertions.

We added the analysis of physical interaction data in the Materials and methods:

“Physical interaction data for *S. cerevisiae* S288c were retrieved from the BioGRID Multi-Validated Physical interaction dataset version 3.5.188 (Oughtred et al., 2019) and the interactome was built using Networkx.”

We analyzed the distributions of physical distance and relative orientation between Ty insertions and neighboring genes and show that differences among lineages are not significantly different. We added one sentence to the fifth paragraph of this section:

“Distributions of physical distance to neighboring genes were similar across lineages, as were distributions of relative orientation (Figure 5—figure supplement 10).”

We added Figure 5—figure supplement 10 to present the results of this analysis.

Former Figure 5—figure supplement 7 was changed to Figure 5—figure supplement 11.

Reviewer #2:[…]1) Did the authors were too conservative with their hybrid design and crossed populations that were too similar? I guess their rational is to mimic what they isolated in nature, but maybe if they look at the *S. cerevisiae* x *S. paradoxus* crosses that they published before, this might change.

We agree with the reviewer that restricting our analyses to the intraspecific divergence range excludes a vast range of higher divergence levels in interspecific crosses. The reason why we initially excluded the data from our *S. paradoxus* x *S. cerevisiae* crosses is that we judged it was less relevant to our natural system of *S. paradoxus* lineages (wild *S. cerevisiae* being largely absent at the latitudes where the analyzed strains were collected). While the choice to focus on intraspecific *S. paradoxus* crosses may seem conservative, we want to highlight their relevance in uncovering the mechanisms that contribute to early isolation in the process of speciation. Nevertheless, we agree that it makes sense to include our *S. paradoxus x S. cerevisiae* crosses in our analyses, which we named BSc in this manuscript. We updated the Introduction text to account for this new category of crosses:

“Here, we use population genomic data and laboratory evolution experiments on *S. paradoxus* and its sibling species *S. cerevisiae* to investigate the hybrid reactivation hypothesis and, more generally, the factors governing TE accumulation in natural and experimental lineages.”

In the first paragraph of this section, we changed the following:

“To systematically explore the effect of hybridization on Ty transposition rates, we performed a large-scale mutation accumulation (MA) experiment on 11 artificial hybrid backgrounds, eight of which were previously published (Charron et al., 2019).”

”Pairs of haploid derivatives of natural *S. paradoxus* and *S. cerevisiae* strains were chosen to span a range of evolutionary divergence (Figure 6B), yielding four categories of intraspecific crosses (from the least to the most divergent): *SpC*×*SpC* (CC), *SpB*×*SpB* (BB), *SpB*×*SpC* (BC) and *SpB*×*SpA* (BA); and one category of interspecific crosses: *SpB*×*SpA* (BSc; Figure 6C, Supplementary file 1A-F).”

We changed the third paragraph of this section as follows:

“Although these models explain little global variance (R^2^=0.40, 0.08 and 0.06 for Ty1, Ty3p and Tsu4 respectively), we found that neither independent variable was a significant predictor of ΔCN (Figure 6F). […] Crosses that shared the *SpB* parental strain UWOPS-91-202 (BB2, BC2, BA2, BSc2) seemed to have systematically higher Ty1 ΔCN values than the corresponding crosses that shared the *SpB* parental strain MSH-604 (BB1, BC1, BA1, BSc1).”

We modified the fourth paragraph of the Discussion as follows:

“This trend is especially important for Ty3p, as eight out of ten genotypes exhibit a negative change in median CN.”

We modified Figure 6 and Figure 6—figure supplement 1.

2) I think the natural strains described in the first part of the paper and used for the crossing, would benefit for characterising TE activity, e.g. looking expression or better using existing reporters that enable to calculate rates of retrotransposition and excision. I think this can have major impact in the results obtained from the MALs and is a more direct read out of the hybridisation effect.

We thank the reviewer for the suggestion of complementing our MA experiments with reporter assays that measure retrotransposition rate. We think this would be an entirely valuable approach to the question, as shown by Smukowski Heil and collaborators (Smukowski Heil et al., 2020), who recently demonstrated that no Ty reactivation occurs in interspecific hybrids between *S. cerevisiae* and *S. uvarum*.

As we understand the reviewer's comment, the occurrence of genomic changes during MA following a hypothetical initial and transient reactivation of retrotransposition might be the major factor making reporter assays a “more direct read out” of retrotransposition rates compared to MA. However, we would like to argue that in the time scale of our MA experiment (less than 800 mitotic generations), and given that we sequenced the strains at the initial timepoint of the MA experiment, it appears unlikely that signatures of an early and transient reactivation would have been purged from the genomes we sequenced. While we agree that many types of genomic changes can yield variation in Ty CNs that is unrelated to retrotransposition *per se*, such a “masking” of reactivation would require a strong mutational bias towards elimination of genomic loci containing Ty elements, or systematically very strong deleterious effects associated with novel Ty insertions that would drive their counter-selection in a MA regime characterized by minimal *N_e_*. We cannot find any good reason to expect any of these scenarios. As we mention in the discussion of the manuscript, we believe that confident assessment of the Ty insertion loci in the MA lines would be essential in gaining further insight into the genomic processes involved in Ty CN change, but we believe it would require new data collection and thorough analyses that are beyond the scope of this manuscript.

We agree that reporter assays are an attractive and certainly powerful way of characterizing transposition rates. However, we believe that many limitations make them hard to apply to the current study in a straightforward and meaningful way. First, to our knowledge, all the retrotransposition marker constructs available are based on a Ty1 copy from *S. cerevisiae*. While this choice is certainly defendable in the case of interspecific hybrids involving *S. cerevisiae* as a parent (as in the case of Smukowski Heil and collaborators and our BSc crosses), most of our results are on intraspecific hybrids of *S. paradoxus*. The large sequence divergence level that separates these species is likely to have functional consequences on the ability of the marker to retrotranspose and impact retrotransposition rates in an unpredictable way, thus prompting the adaptation of these tools to *S. paradoxus* Ty elements. Moreover, MA experiments offer the advantage of exploring retrotransposition activity of all families at the time. Thus, exploring the dynamics of all families would ideally require generating constructs that account for both inter-lineage and intra-strain variation in Ty content. Second, while its interpretation might seem straightforward at first sight, retrotransposition marker assays require the function of many cellular processes that can exhibit phenotypic diversity at the intraspecific level and thus need to be carefully controlled for. In order to yield sufficiently high numbers of mutants to reach appreciable sensitivity in a reasonable experimental setup, most assays require transcriptional induction using the *GAL1* promoter. In *S. cerevisiae*, it was shown that very low levels of intraspecific genetic variation results in wide phenotypic variation in the regulation of galactose genes (Lee et al., 2017). Additionally, splicing efficiency of the artificial introns inserted into *HIS3* or GFP can also exhibit intraspecific variation. Ruling out potential effects from these confounding processes requires many controls in the form of assays and constructs that are not readily available to us. Finally, our choice of experimental condition for the MA experiment (YPD) is incompatible with most reporter assays, which require induction in galactose and selection of a plasmid, either by auxotrophy complementation in synthetic medium or antibiotic resistance in complex medium. All these requirements are important deviations from the basic YPD medium used in the MA experiment. Thus, the results from reporter assays would be hard to interpret, as it would be difficult to disentangle the orthogonal experimental approach from environmental effects.

In conclusion, the difficulty to identify a clear drawback to our MA approach given our sequencing strategy, and the complexity of adapting reporter assays and their necessary controls to *S. paradoxus,* prompt us to argue against the inclusion of reporter assays as a revision to our manuscript. While this is certainly a valid and important orthogonal approach to our question, we believe that its complexity deserves to be addressed in a separate future study.